

# Advanced CFD-MBS coupling to assess low-frequency emissions from wind turbines

Levin Klein[1], Jonas Gude[1], Florian Wenz[1], Thorsten Lutz[1], and Ewald Krämer[1]

[1]Institute of Aerodynamics and Gas Dynamics, University of Stuttgart, Pfaffenwaldring 21, 70569 Stuttgart, Germany

**Correspondence:** Levin Klein (levin.klein@iag.uni-stuttgart.de)

**Abstract.** The low-frequency emissions from a generic 5 MW turbine are investigated numerically. In order to regard airborne noise and structure-borne noise simultaneously a process chain was developed. It considers fluid-structure coupling (FSC) of a computational fluid dynamics (CFD) solver and multibody simulations (MBS) solver as well as a Ffowcs Williams-Hawkings (FW-H) acoustic solver. The approach was applied to a generic 5 MW turbine to get more insight into the sources and mechanisms of low-frequency emissions from wind turbines. For this purpose simulations with increasing complexity in terms of considered components in the CFD model, degrees of freedom in the structural model and inflow in the CFD model were conducted. Consistent with literature, it has been found that aeroacoustic low-frequency emission is dominated by the blade-passing frequency harmonics. The tower base loads, which excite seismic emission, tend to be dominated by structural eigenfrequencies with increasing complexity of the model. The main source of aeroacoustic emissions is the blade-tower interaction and the contribution of the tower as an acoustic emitter is stronger than the contribution of the rotor. Aerodynamic tower loads also significantly contribute to the external excitation acting on the structure of the wind turbine.

## 1 Introduction

Renewable sources of energy and especially wind power have seen a strong expansion in the last years. Even though the construction of large offshore wind farms is currently a strong focus, the potential of onshore wind turbines by opening up new, previously unused areas and repowering of existing sites is still significant. With regard to the acceptance and the fulfillment of stricter legal requirements concerning noise and vibrations, the research on low-frequency emissions from wind turbines gains importance. As wind turbines are counted among the tallest machines on the planet that work in an uncontrolled outside environment, noise and vibration emissions in a broad frequency range occur. While sources of acoustic wind turbine emission in the audible range are widely researched, understood and different methods are applied to reduce aerodynamic and mechanical noise (Liu, 2017), there is much less known about low-frequency emissions from wind turbines. Many publications about low-frequency emissions of wind turbines concentrate on the impact on seismic measurements. The emitted ground motion signals from wind turbines are measured by local seismic stations built for detection of events with small magnitudes like far away earthquakes or nuclear weapons tests. Zieger and Ritter (2018) observed an increase of amplitudes in a frequency range from 0.5 Hz to 10 Hz dependent of the rotational speed of the turbine and thus wind speed at a distance of 5.5 km away from a wind turbine. This confirms the measurements by Stammler and Ceranna (2016) and Styles et al. (2005) who found that nearby





wind turbines reduce the sensitivity of seismic stations as they introduce wind dependence into the measured noise spectra. Acoustic measurements in the low-frequency range $3.3\,\text{km}$ from a wind farm show discrete peaks at the blade-passing frequency (BPF) and its higher harmonics below $20\,\text{Hz}$ (Hansen et al., 2017). This was also observed by Pilger and Ceranna (2017) who evaluated the data obtained by a microbarometer array for infrasound detection located in northern Germany. Zajamšek et al. (2016) investigated the measurability of these acoustic waves in buildings. Hence, the blade-tower interaction is seen to be responsible for aeroacoustic low-frequency noise of windfarms (Van den Berg, 2005).

Knopper et al. (2014) conclude from their literature survey that human health is not likely to be affected by low-frequency noise and infrasound from wind turbines.Turnbull et al. (2012) state that the measured level of infrasound within two Australian wind farms was similar to that measured in urban and coastal areas and near other engineered noise sources.

For an optimization of the structure and foundations of future wind turbines as well as for the assessment of the impact of low-frequency noise and low-frequency seismic vibrations on the environment, reliable methods for the prediction of emissions are of great importance. Gortsas et al. (2017) performed a numerical study to calculate wave propagation using the Boundary Element Method. They developed a model which considers the mentioned seismic vibrations as well as the low-frequency noise in air and even allows a prediction of the sound pressure level (SPL) inside a generic building. But, as this model is only capable to calculate the propagation, reliable input data representing the airborne and structure-borne emissions from the wind turbine has to be provided. CFD simulations including fluid-structure interaction (FSI) are capable of providing both. Thus, Gortsas et al. used data made available by the authors of the present paper.

There are few studies on the modelling of aeroacoustic low-frequency emission from wind turbines. In the 1980s the NASA developed a code for predicting low-frequency wind turbine noise based on Lowson's acoustic equation applied on rotor forces (Viterna, 1981). Madsen (2010) presented a Blade Element Momentum (BEM) based investigation of low-frequency noise that uses the same theory for the aeroacoustic model. CFD simulations combined with the FW-H propagation method have been applied by Ghasemian and Nejat (2015) and Bozorgi et al. (2018) to assess low-frequency noise of wind turbine rotors. While Madsen considers the influence of the tower on the rotor aerodynamics, Ghasemian and Nejat and Bozorgi et al. study the isolated rotor. Yauwenas et al. (2017) investigated the blade-passage noise of a generic model turbine numerically using CFD and Curle's acoustic analogy. They found a significant contribution of the induced pressure fluctuations on the tower to the tonal blade-passage noise which was validated with experimental measurements.

In recent years, CFD based fluid-structure coupling has been applied frequently for the investigation of wind turbines. Li et al. (2017) presented a framework of a wind turbine aero-servo-elastic simulation including flexible blades and tower which allows motion of all turbine components. In his approach, controllers for torque and blade pitch are included as well and he focuses his studies on the impact of FSI on aerodynamic rotor loads, drive train dynamics, controllers and wake. Streiner et al. (2008) developed a coupling of the CFD code *FLOWer* to the multibody solver (MBS) *SIMPACK* with the capability to couple isolated wind turbine rotors. A totally new *FLOWer-SIMPACK* coupling is revealed in the present paper with the potential to take into account more degrees of freedom, like tower deformation or changes in rotational speed in the structural model and their impact on aerodynamics and aeroacoustics, respectively. Together with the already existing process chain, fully coupled CFD



simulations under realistic turbulent inflow conditions can be conducted, providing both airborne and structure-borne emissions simultaneously. A FW-H in-house code is applied to calculate aeroacoustic immission at distant observers while tower base loads represent the structure-borne emission. The aim of the present paper is to identify the sources of low-frequency emissions and to investigate the impact of the complexity of the numerical model on the calculated low-frequency emissions

from a generic $5\,\mathrm{MW}$ wind turbine. The complexity of the model was increased from a rotor only simulation with uniform inflow to a coupled simulation including blade, tower and foundation dynamics with turbulent atmospheric boundary layer. The spectra of tower base loads and acoustic immissions for overall 7 cases were compared in a frequency range from $0.1$ to $25\,\mathrm{Hz}$ for evaluation.

## 2   Numerical process chain

A high fidelity process chain based on multiple solvers was established for the investigation of low-frequency emissions from wind turbines. It consists of the CFD solver *FLOWer*, the MBS solver *SIMPACK* and the FW-H solver *ACCO*. A strong coupling between *FLOWer* and *SIMPACK* was developed to generate high fidelity time series of surface pressure distribution on the turbine and structural loads (forces and moments) acting on the foundation of the turbine. Using the CFD results, the aeroacoustic signal at distant, predefined observer positions is computed by means of *ACCO*.

### 2.1   CFD solver

*FLOWer* is a compressible, dual time stepping, block structured Reynolds-averaged Navier-Stokes (RANS) solver developed by German Aerospace Center (DLR) (Kroll et al., 2000). The usage of independent grids for bodies and background is enabled by the overlapping grid technique *CHIMERA*, one of *FLOWers* main features. The solver is continuously extended at Institute of Aerodynamic and Gas Dynamics (IAG) regarding functionality and performance, including, amongst others, the higher order

finite difference weighted essentially non-oscillatory (WENO) scheme (Kowarsch et al., 2013), Dirichlet boundary condition to apply arbitrary unsteady inflow, a body forces approach to superimpose turbulence (Schulz et al., 2016b) and various DES schemes (Weihing et al., 2016). The capability of *FLOWer* for wind turbine simulations has been shown in several projects. The interaction of a wind turbine in complex terrain with atmospheric turbulence was investigated by Schulz et al. (2016a) and code to code comparisons were recently conducted in the European *AVATAR* project (Schepers et al., 2016).

### 2.2   Multibody solver

*SIMPACK* is a commercial non-linear MBS solver that can be applied to simulate dynamic systems consisting of rigid and flexible bodies. Flexible turbine components like tower and blades are modeled with linear or nonlinear beam theory. The kinematics between the components are defined by joint elements and internal forces can be considered. There are two ways to

apply external forces such as aerodynamic forces, either by built-in interfaces or by programmable user routines. Controllers



can also be integrated. *SIMPACK* is recently applied by industry and research groups for the simulation of wind turbines, examples can be found in (Luhmann et al., 2017; Jassmann et al., 2014).

## 2.3 Fluid-structure interaction

To take the influence of unsteady structural deformation on the aerodynamics into account, a coupling between *FLOWer* and *SIMPACK* was implemented. The new approach generally allows coupling of slender beam like structures and is not limited to rotor blades or even wind turbines. Combined coupling of rotating and non-rotating parts can be applied and the deformation of adjacent structures is considered. Furthermore, coupling is not restricted to flexible deformations but also rigid body motions (rotations and translations) can be realized. In the application of wind turbines e.g. pitch motions and changes in rotational speed of the rotor can be transferred from the MBS solver to the CFD solver.

For the technical realization, an existing interface that was developed to couple *SIMPACK* with the fluid solver *ANSYS CFX* for the investigation of a tidal current turbine (Arnold et al., 2013) was extended. Furthermore, libraries for grid deformation and load integration which were recently developed and integrated into *FLOWer* (Schuff et al., 2014; Kranzinger et al., 2016) had to be extended for the coupling with *SIMPACK*. Besides the functionality, the main target of the implementation was to keep the set-up of the coupling simple and the dependencies between MBS and CFD models low. Thus, resolution of CFD and MBS model are independent of each other which allows a fast and easy adjustment and replacement of MBS structures or CFD meshes. Furthermore, the new coupling can be restarted, allowing much longer simulation times if *FLOWer* runs on clusters with limited job duration. It was already successfully applied on the blade of a generic $10MW$ turbine for comparison reasons by Sayed et al. (2016) who implemented a coupling of *FLOWer* to the structural dynamics solver *Carat++*.

### 2.3.1 General functionality

The developed coupling is a partitioned approach, where two independent solvers run simultaneously on different machines and exchange data via Secure Shell (SSH) connection at discrete positions, so called markers. The markers are positioned inside the bodies. While rigid bodies have only one marker, flexible bodies like rotor blades have several markers that are distributed along the beam. On the one hand, deflections and rotations of these markers relative to their non-deformed position are computed by *SIMPACK*. On the other hand, aerodynamic forces and moments acting on these markers are calculated in *FLOWer*. For each structure that is coupled, a communication coordinate system is defined that has to be in the same position and same orientation in both models at all times. It does not have to be fixed, but can be rotating or translating in a predefined way. All data concerning the respective structure is communicated in this coordinate system.

### 2.3.2 Mesh deformation

The task of the deformation library implemented in *FLOWer* is to apply the deformations of the markers on the corresponding CFD surfaces and to deform the surrounding volume mesh accordingly. The surface is represented by a point cloud which is generated from the CFD mesh. For rigid structures only one marker is used and all surface cloud points perform a rigid





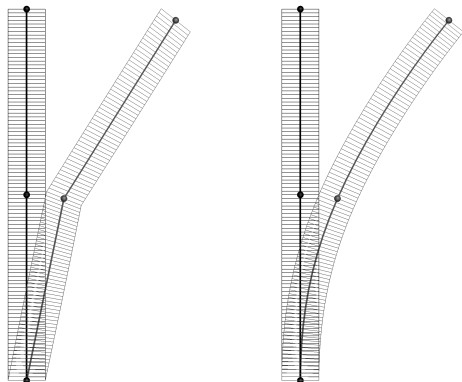

**Figure 1.** Undeformed (black) and deformed (grey) surface mesh, linear interpolation (left) and spline interpolation (right) for a simple test case with three markers. Generic deformation of first bending mode. Rotation at lower end is zero.

body motion based on the translation and rotation of this marker. A cubic spline interpolation is applied for the mapping of flexible structures (beams) consisting of more than one marker. The deformation of each surface cloud point is then realized as rigid body motion based on the corresponding positions along the beam. While a complete spline approach is used for the deflections, taking the rotation at the end points into account, the rotations and the non-deformed marker positions are

5   interpolated using natural splines. A similar approach has been presented by Arnold et al. (2013). Figure 1 shows the surface grid deformation for the first bending mode in a simple test case with 3 markers. Spline interpolation gives a much smoother result in comparison to linear interpolation and considers the non-rotated lower end.

Finally, the volume grids are deformed based on the deformation of the point cloud using radial basis functions. To ensure correct overlapping of deformed meshes, holes associated to the deformed surface can also be deformed.

### 2.3.3   Load integration

The load library implemented in *FLOWer* enables the calculation of aerodynamic loads on grid surfaces by integration of friction and pressure over the cell faces. For the coupling to *SIMPACK*, the CFD surface is divided into segments based on the deformed marker positions. Loads are integrated for these segments and assigned to the respective markers. Moments are calculated with respect to the origin of the corresponding communication coordinate system. For structures with only one

15   marker, loads are integrated over the whole CFD surface of the respective structure.

### 2.3.4   Communication interface

The communication is realized by means of files. Data files contain deformations or loads and status files indicate that the data file is ready to be read. While *SIMPACK* is running on a local Windows machine, *FLOWer* is usually executed in parallel mode on a high performance computing (HPC) system running on Linux. A portable communication script in Windows in-





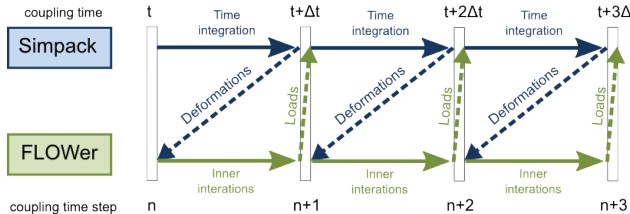

**Figure 2.** Explicit coupling scheme of the *FLOWer-SIMPACK* coupling.

herent scripting language PowerShell enables fast and reliable communication between the two solvers. The Linux machine is accessed using a SSH connection via the Windows Secure Copy (WinSCP) client.

### 2.3.5   Coupling scheme

In the presented work, an explicit coupling scheme was applied. The size of the coupling time step is equal to the physical
*FLOWer* time step and remains constant throughout the simulation. Both solvers are running in a sequential way, waiting for the other solver to reach the next time step and to send communication data. *SIMPACK* is running one time step ahead doing time integration with the aerodynamic loads that *FLOWer* computed at the end of the previous time step (Figure 2).

### 2.4   Acoustic solver

Acoustic immission at arbitrary observer locations was calculated by means of the in-house FW-H solver *ACCO*. Pressure
and velocities on surfaces enclosing the noise sources are evaluated at each time step of the transient CFD solution, including velocities due to deformation, translation and rotation. For the present study, the surfaces used for the acoustic analysis were identical with the physical surfaces of the turbine (rotor, tower, hub etc.). Volume sources generated by free-flow turbulence were neglected, which is justified for low mach number flow because quadrupole volume noise is proportional to $Ma^7$. This approach was validated for a rod-cylinder configuration and an airfoil in turbulent flow (Lutz et al., 2015; Illg et al., 2015). The
acoustic monopole and dipole contributions to the observer sound pressure level (SPL) are computed by means of the Ffowcs Williams-Hawkings (FW-H) equation. Its left-hand side is the wave equation which describes the transmission of sound to the observer, presuming undistorbed propagation and observers located in the acoustic far field. Hence, ground reflections and non-linear propagation due to atmospheric layering and turbulence are not taken into account. The acoustic far field is defined by the presence of a fully developed wave front and thus starts several wave lengths away from the source. Parallel execution
of *ACCO* allows the computation of noise carpets consisting of several thousand observer locations.

The application of the FW-H analogy allows evaluation of the contribution of selected components of the wind turbine to SPL by excluding surfaces of particular components (e.g. tower) from the analysis.





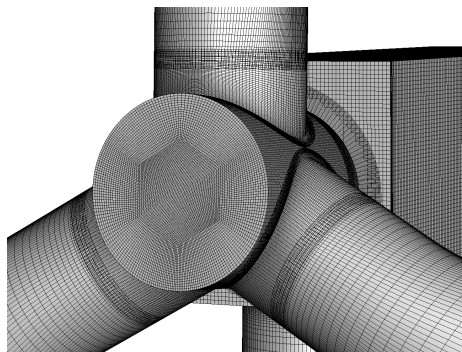

**Figure 3.** CFD surface mesh, showing the connection of hub, blades and nacelle with overlapping meshes.

## 2.5 Computational set-up

### 2.5.1 The turbine

The examined turbine is based on the NREL $5\,\text{MW}$ turbine (Jonkman et al., 2009) and was slightly modified in the *OFFWINDTECH* project (Bekiropoulos et al., 2013). The main modifications concern the rated conditions which were changed to a rotational

speed of $11.7\,\text{RPM}$ and a pitch angle of $-2.29°$ at a wind speed at hub height of $11.3\,\text{ms}^{-1}$. The turbine was investigated at rated conditions in an onshore configuration with a hub height of $90\,\text{m}$, a rotor diameter of $126\text{m}$ with a tilt angle of $5°$ and a precone angle of $2.5°$. The original tower with a bottom diameter of $6\,\text{m}$ and a top diameter of $3.87\,\text{m}$ was used.

### 2.5.2 CFD model

The CFD model of the *OFFWINDTECH* turbine consists of ten independent body meshes, that are embedded in a Cartesian

hanging grid node background mesh using the *CHIMERA* technique. Blades, hub, nacelle and tower were considered in the simulation with fully resolved boundary layer ($y^+ \leq 1$). No gaps are left between the components of the turbine, as blade-hub connectors and a hub-nacelle connector are included in the CFD mesh (Figure 3). Blades were meshed in a C-H-mesh topology with 120 cells in radial direction and 180 cells around the airfoil, summing up to approximately 5.3 million cells per blade. Two different Cartesian background grids were created using hanging grid nodes. One for the case with prescribed

atmospheric turbulence where the mesh is additionally refined to a cell size of $1\text{m}^3$ upstream of the turbine (64.5 million cells) and another for the case without atmospheric turbulence where only the mesh close to the turbine is refined (20.8 million cells). The computational domain is approximately $48.8$ rotor radii $(R)$ long ($12.7\,R$ upstream of the rotor plane), approximately $24.4\,R$ wide and has a height of approximately $16.2R$. According to a previous study using *FLOWer* (Sayed et al., 2015), the background grids were expanded more than sufficient in all directions to avoid influence on the flow field around the turbine.

Overall the two set-ups consist of 86 million (fine) respectively 42 million cells (coarse).

Concerning inflow three different cases are regarded in the present study. Uniform inflow, steady atmospheric boundary layer and turbulent atmospheric boundary layer. An exponent of $0.19$ was applied for the power law profile describing the steady



**Table 1.** Details on the foundation of the wind turbine, similar to Gortsas et al. (2017).

| Mass | $1.888e6\,\mathrm{kg}$ |
|---|---|
| Inertia $x,y$ | $82.705e6\,\mathrm{kgm^2}$ |
| Inertia $z$ | $88.529e6\,\mathrm{kgm^2}$ |
| Stiffness $x,y$ | $8.554e9\,\mathrm{Nm^{-1}}$ |
| Stiffness $z$ | $7.332e9\,\mathrm{Nm^{-1}}$ |
| Rotational stiffness $x,y$ | $559e9\,\mathrm{Nm\cdot rad^{-1}}$ |
| Rotational stiffness $z$ | $559e9\,\mathrm{Nm\cdot rad^{-1}}$ |
| Damping $x,y$ | $240e6\,\mathrm{Nsm^{-1}}$ |
| Damping $z$ | $325e6\,\mathrm{Nsm^{-1}}$ |
| Rotational damping $x,y$ | $5.035e9\,\mathrm{Nms\cdot rad^{-1}}$ |
| Rotational damping $z$ | $4.180e9\,\mathrm{Nms\cdot rad^{-1}}$ |

atmospheric boundary layer, keeping the wind speed at hub height at $11.3\,\mathrm{ms^{-1}}$. Atmospheric turbulence with a reference length scale of $42\,\mathrm{m}$ was created using Mann's model (Mann, 1994) and introduced into the flow field using body forces $16\,\mathrm{m}$ downstream of the inlet, superimposing the steady boundary layer profile. The resulting turbulence level at the turbine position was $16\%$. Unsteady RANS (URANS) simulations were applied with a second order dual time stepping scheme for temporal

discretisation. The second order central discretisation with the Jameson-Schmidt-Turkel (JST) artificial dissipation term was used for spatial discretisation in body meshes and fifth order WENO scheme was applied on the background mesh in order to reduce dissipation of vortices. Menter-SST (Menter, 1994) was deployed for turbulence modelling. A physical time step corresponding to $0.75°$ azimuth ($\approx 0.0168\,\mathrm{s}$) with 100 inner iterations was applied for the evaluated part of the simulations.

### 2.5.3   Structural model

The *SIMPACK* model of the *OFFWINDTECH* turbine was built by Matha et al. (2010). The blades are modelled non-linear by using multiple flexible bodies per blade. The structural properties of the tower are adopted from the NREL $5\,\mathrm{MW}$ turbine (Jonkman et al., 2009) taking 20 modes into account. Hub and nacelle are defined as rigid bodies. The foundation is modelled as rigid body connected to the ground with a spring-damper system. Detail can be found in Table 1.

### 2.5.4   FSI setup

The coupling between *FLOWer* and *SIMPACK* for the *OFFWINDTECH* turbine was applied using 160 markers (Figure 4), 49 markers for each blade, 11 markers for the tower, and nacelle and hub with one marker each. Since in the structural model and the CFD model a fixed rotational speed was prescribed, a rotating communication coordinate system in the center of the hub was used for the rotating parts. The communication for tower and nacelle was performed in a fixed coordinate system placed at the tower base (Figure 4). In the *SIMPACK* model of the turbine, additional rigid bodies were created for the definition of



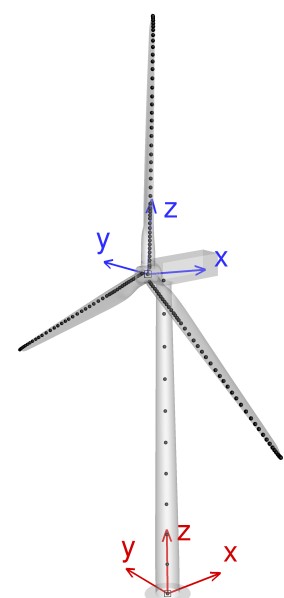

**Figure 4.** CFD surface of turbine including markers for coupling with *SIMPACK*. Rotating hub coordinate system is shown in blue and tower base coordinate system in red.

**Table 2.** Definition of simulation cases, ordered with increasing complexity.

| Case name | Inflow | CFD structures | Flexible structures | Background mesh |
|---|---|---|---|---|
| LC1 | uniform | rotor | none | coarse |
| LC2 | uniform | rotor, nacelle, tower | none | coarse |
| LC2_FSC1SD | uniform | rotor, nacelle, tower | rotor blades SD | coarse |
| LC2_FSC1 | uniform | rotor, nacelle, tower | rotor blades | coarse |
| LC2_FSC3 | uniform | rotor, nacelle, tower | rotor, nacelle, tower, foundation | coarse |
| LC3_FSC3 | steady ABL | rotor, nacelle, tower | rotor, nacelle, tower, foundation | fine |
| LC4_FSC3 | turbulent ABL | rotor, nacelle, tower | rotor, nacelle, tower, foundation | fine |

ABL, atmospheric boundary layer; SD, steady deformation.

the undeformed markers. The corresponding moving markers were attached to the flexible structures of the turbine. With this approach the measured deformations between deformed and undeformed markers are composed of flexible deformations of the body itself plus rigid body motion due to deformation or motion of the adjacent body.





### 2.5.5 Simulation cases

In Table 2 all regarded simulation cases are listed. Three studies were conducted. In the first study, no FSI was considered and thus all turbine components were kept rigid. The influences of the presence of the tower and the distance of the blade to the tower were evaluated at uniform inflow conditions by comparing LC1, LC2 and LC2_FSC1SD. In case LC2_FSC1SD the averaged blade deformations of case LC2_FSC1 were used to create a realistically deformed shape of the blades with reduced distance between blades and tower. In a second study the degrees of freedom of the structural model were increased at uniform inflow conditions. Three cases were compared: a rigid case with steady deformed blades (LC2_FSC1SD), a case with flexible blades (LC2_FSC1) and a case with flexible blades as well as a flexible tower and foundation (LC2_FSC3). In the third study, the inflow conditions were changed, keeping the structural model the same. Case LC2_FSC3 is used as reference. A steady atmospheric boundary layer (ABL) was prescribed at the inlet by means of a power law inflow profile in case LC3_FSC3. This steady ABL was superposed with velocity fluctuations modelling a turbulent atmospheric boundary layer in case LC4_FSC3.

### 2.5.6 Computational approach

One feature of the implemented coupling is that coupled simulations can be started from results of standalone CFD simulations. This was applied in the presented research to achieve a well converged state concerning aerodynamic forces and flow field. At the same time computational costs could be saved, as coupled simulations with various degrees of freedom could be started from the same converged state. In all cases at least 32 revolutions were simulated before the start of the coupling. This was necessary due to the high induction of the rotor. The structural simulation is started from a initialized state at the beginning of the coupling. All flexible components are released from a rigid state and due to the sudden impact of gravitational, centrifugal and aerodynamic forces, deformations tend to overshoot. As the CFD part of the coupled simulations is computationally expensive, it is important to have a fast convergence of deformations and loads to a periodic state. While flap-wise deflections of the blades are damped very fast, blade edge-wise deflections and tower deflections are not. *SIMPACK* allows the user to define time depended functions for external forces and dampers. As a first step, to reduce deformation velocity at the start of the coupling, aerodynamic loads are multiplied with a linearly increasing load factor over the first $120$ time steps ($\approx 1,71\,\mathrm{s}$). Additionally, dampers in form of counter acting forces proportional to deformation velocity are attached to the tower tip as well as to the blades to damp initial oscillations of blades and tower. The damping factors are linearly increased and decreased over time and were determined manually for optimal performance of the model. Figure 5 exemplarily shows the force of the tower tip damper in $x$-direction and the deflection of the tower tip for case LC2_FSC3. The tower damping factor decreases to zero after $740$ time steps ($\approx 7.91\,\mathrm{s}$). The tower tip deflection shows only a small overshoot and is well converged when the damper is switched off. With this approach, a fast convergence of deflections and loads was achieved and only the first two coupled revolutions of the turbine could not be used for evaluation.



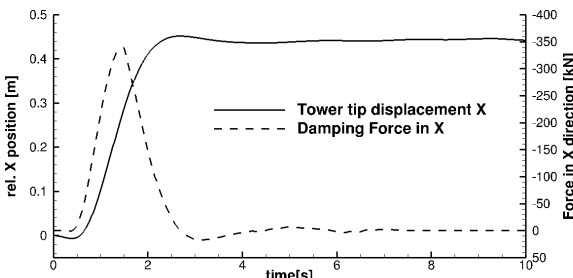

**Figure 5.** Tower tip deflection and artificial damping force on tower tip in $x$-direction at start of coupled simulation.

## 2.6 Evaluation

The aim of the simulation chain is to model airborne and structure-borne emissions simultaneously by evaluating acoustic immission at distant observers and load fluctuations at the tower base. In the fluid-structure coupled simulations tower base loads were evaluated directly in the structural model at the interface between tower and foundation, whereas in the non-coupled simulations aerodynamic loads were computed from CFD results. In both cases the tower base loads are presented with respect to the tower base coordinate system which is shown in Figure 4. The temporal resolution of the data is equal to the coupling time step. To achieve the same temporal resolution in the acoustic emission, each time step a CFD surface solution was saved as input for the acoustic simulations.

Acoustic simulations using *ACCO* were conducted to calculate the immission at a carpet of observers on the ground surrounding the turbine. Figure 6 shows the 3600 observers located on 20 concentric rings around the turbine at radial positions of $100\,\mathrm{m}$ to $2000\,\mathrm{m}$ with a radial resolution of $100\,\mathrm{m}$ and a circumferential resolution of $2°$. Unweighted SPL was calculated from sound pressure time series at the observers with a reference sound pressure of $20\,\mu\mathrm{Pa}$. The sound propagation and directivity for discrete frequencies can be evaluated by plotting the SPL contour on the ground. Four observers at a distance of $1000\,\mathrm{m}$ to the turbine were chosen for detailed evaluation of SPL spectra (large dots in Figure 6). Prior to frequency analyses by means of fast Fourier transform (FFT), the time series signals of loads and sound pressure were cut to multiples of one rotational period of the turbine in order to supply a preferably periodical signal to the FFT and to avoid influence of start-up effects. In coupled simulations, the first two revolutions were excluded from evaluation. For case LC4_FSC3 14 revolutions and for all other cases 8 revolutions were evaluated.

## 3 Results

### 3.1 Rigid simulations

In this section three non-fluid-structure coupled cases are compared at uniform inflow conditions. As reference the rotor only case (LC1) is regarded where unsteady effects on the loads only result from the tilt of the rotor, the proximity to the ground





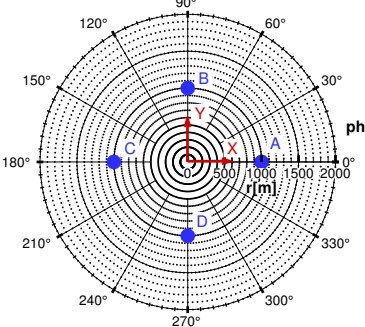

**Figure 6.** Observer positions for evaluation of aero acoustic emissions. Tower base coordinate system shown in red. View from above, turbine in the center, wind from left.

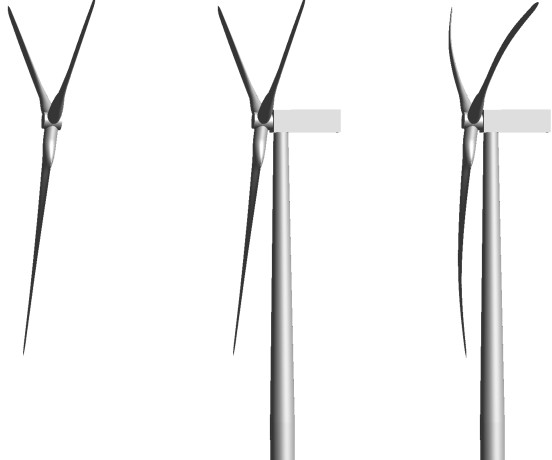

**Figure 7.** CFD turbine surfaces of cases LC1 (left), LC2 (middle) and LC2_FSC1SD (right). Snapshot with one blade in front of the tower at $180°$ azimuth.

and unsteady flow separation. In a second case, the tower is considered (LC2) and in a third case steady deformation is applied to the blades (LC2_FSC1SD). The CFD surfaces of all three cases are shown in Figure 7.

### 3.1.1 Tower base loads

In the non-fluid-structure coupled cases no unsteady structural forces occur as all structures are rigid. Thus, load fluctuation
5  only arise from aerodynamics. Figure 8 shows the spectra of the aerodynamic loads of all three cases with respect to the tower base coordinate system. No distinctive peaks can be found in the spectra of LC1. After including the tower in the simulation (LC2), sharp peaks at the blade-passing frequency and its higher harmonics appear with significantly increased amplitudes up to a frequency of approximately $10\,\mathrm{Hz}$. Regarding $F_y$ and $M_x$, a general increase of the amplitudes below BPF are present with



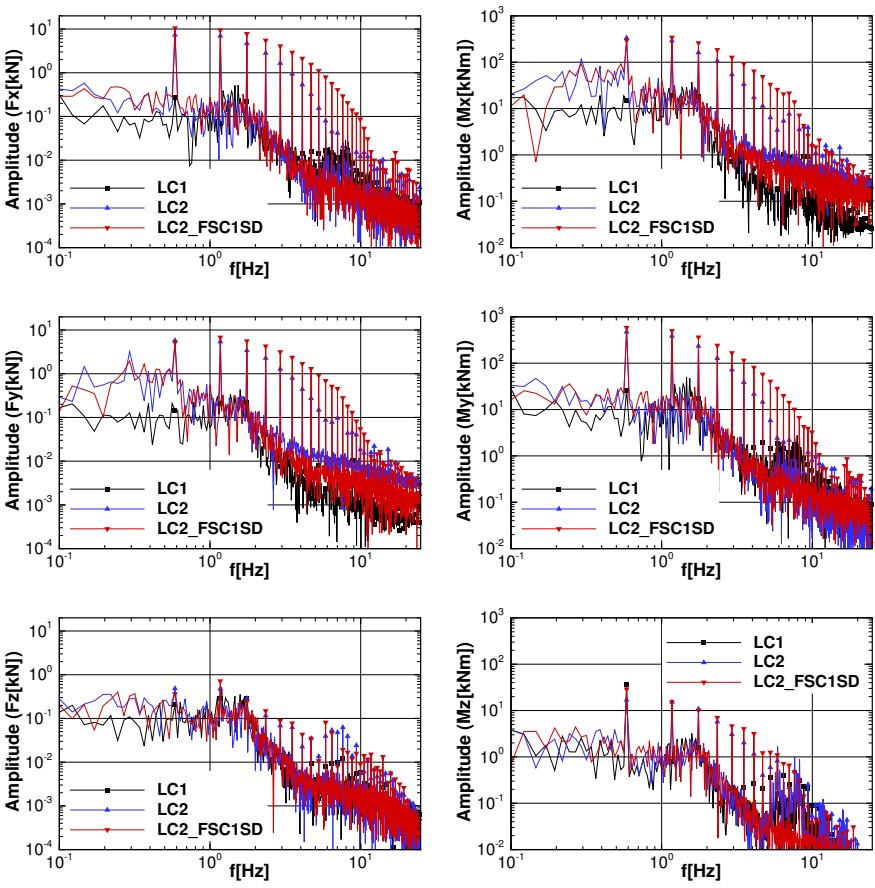

**Figure 8.** Spectra of tower base loads for cases LC1, LC2 and LC2_FSC1SD.

a peak at approximately $0.3\,\text{Hz}$ caused by vortex shedding, which will be shown later. In LC2_FSC2SD the distance between tower and blades is reduced due to the steady deformation of the blades. This leads to an increase of the amplitudes at blade-passing harmonics. The relative increase is stronger for higher frequencies. The amplitude of $F_x$ is increased by more than $50\%$ for frequencies between $5\,\text{Hz}$ and $10\,\text{Hz}$. For $F_y$ and $M_x$ the amplitude at BPF stays almost constant while amplitudes are

5    increased for the higher harmonics compared to case LC2. The maximum amplitude of $M_x$ is shifted to the second harmonic of BPF. The amplitudes of $F_z$ and $M_z$ are much lower compared to the other load components.

The composition of the loads was investigated in detail for case LC2_FSC1SD. Therefore, aerodynamic loads on rotor and tower were evaluated separately. Figure 9 shows the resulting spectra. For all loads except $F_z$ and $M_z$, the peak amplitudes of the tower spectra are dominant over the whole frequency range. Especially for $F_y$ and $M_x$ the tower load amplitudes are

10    up to ten times higher compared to the rotor load amplitudes. For $F_y$ and $M_x$ the general level below BPF is higher in the tower load spectra. This can be interpreted as the impact of unsteady flow separation at the tower induced by vortex shedding. This phenomenon, known as von Kármán vortex street, leads to unsteady forces on blunt bodies with a frequency described



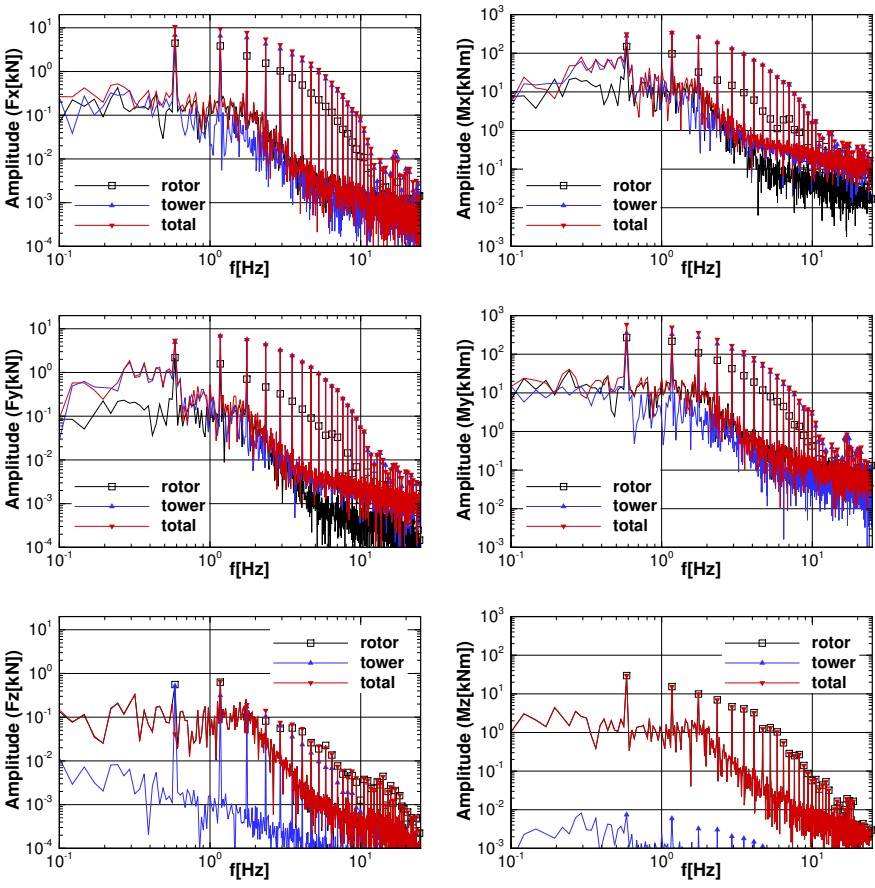

**Figure 9.** Spectra of tower base loads for case LC2_FSC1SD.

by the dimensionless Strouhal number. Assuming a Strouhal number of $0.2$ and an inflow velocity of $8\,\mathrm{ms}^{-1}$ (reduced due to induction of the rotor), the frequency of the undisturbed vortex shedding should be around $0.32\,\mathrm{Hz}$ with respect to the mean diameter of the tower of $4.9\,\mathrm{m}$. As both, diameter and inflow velocity are not constant over the length of the tower and inflow is disturbed by the rotor, a broader range of vortex shedding frequencies can be expected. In Figure 10 the time series of

5 aerodynamic loads $F_y$ and $M_x$ acting on the tower are displayed. It is clearly visible that the peaks appearing periodically with the BPF are superimposed with a lower frequency oscillation. The surface pressure amplitudes on the tower are displayed in Figure 11 at two different frequencies. At BPF ($0.585\,\mathrm{Hz}$) as well as at $0.292\,\mathrm{Hz}$ where the spectra of $F_y$ and $M_x$ have a local maximum. A strong peak appears at BPF at the front of the tower shifted to the side of the approaching blade. The symmetric shape of the pressure amplitude distribution and the higher amplitudes at the rear side of the tower at $0.292\,\mathrm{Hz}$ can

10 very likely be associated with vortex shedding creating the peak in the load spectra. These observations support the idea of the superposition of blade-passing effects and vortex shedding at the tower.



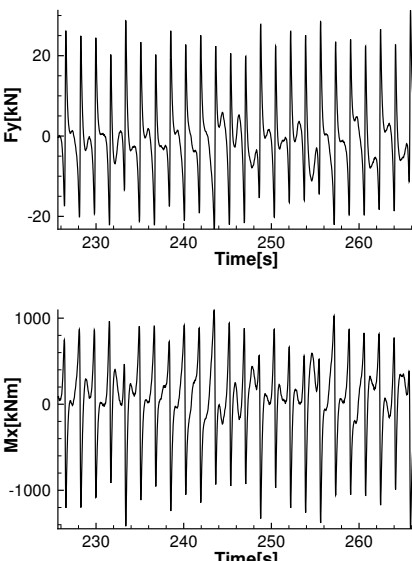

**Figure 10.** Time series of aerodynamic tower loads $F_y$ (top) and $M_x$ (bottom) with respect to the tower base coordinate system for case LC2_FSC1SD.

### 3.1.2 Aeroacoustic emission

Figure 12 shows the spectra of the SPL for observers A-D (Figure 6) for the cases LC1, LC2 and LC2_FSC1SD. The maximum SPL for LC1, the case without tower, occurs at observer B at BPF and is the only prominent peak. The emission at this frequency shows a strong directivity, as the amplitude is much higher at the sides than upstream and downstream of the turbine.
The presence of the tower (LC2 and LC2_FSC1SD) causes a massive increase of amplitudes at the BPF harmonics while the broadband noise level stays low. The highest peak appears upstream of the turbine at observer C at the third BPF harmonic and is approx. $4\,\mathrm{dB}$ higher in case LC2_FSC1SD compared to case LC2. The spectra of case LC2 show only a weak directivity for the BPF harmonics as the amplitudes at the upstream and downstream observers are just slightly lower than at the side observers. A stronger directivity can be observed for case LC2_FSC1SD at BPF where the amplitudes are clearly higher at the upstream and downstream observer. Compared to case LC1 the SPL at frequencies below BPF also rises, but only at observer positions B and D. Comparing LC2_FSC1SD to LC2, the increase of amplitudes due to reduced blade-tower distance is most prominent between fifth and tenth harmonic of BPF where it amounts to more than $10\,\mathrm{dB}$. The SPL peaks drop below $20\,\mathrm{dB}$ at around $15\,\mathrm{Hz}$ even for case LC2_FSC1SD.

To examine the aeroacoustic noise emission in detail, the noise emission originating from tower and rotor surfaces were evaluated separately for case LC2_FSC1SD. Figure 13 shows the SPL spectra at observer positions A-D. It can be seen that for all BPF harmonics the calculated SPL emitted by the tower is higher than the one emitted by the rotor. The global maximum of the rotor induced immission is about $8\,\mathrm{dB}$ lower compared to the global peak of the tower induced immission, both occur



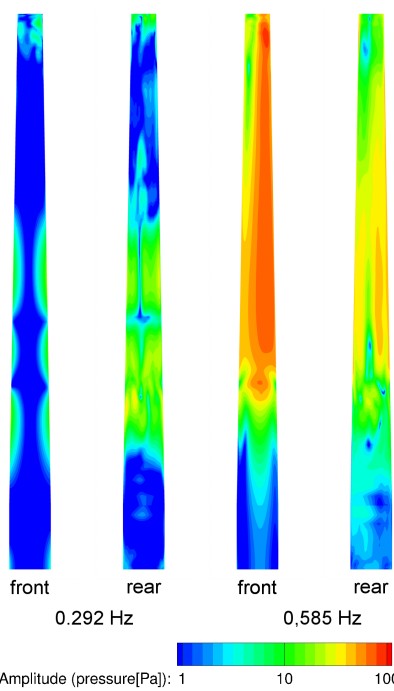

**Figure 11.** Pressure amplitudes on CFD tower surface of case LC2_FSC1SD at $0.292\,\text{Hz}$ (left) and blade-passing frequency ($0.585\,\text{Hz}$) (right).

at observer C. The emission from the rotor shows a strong directivity to the upstream and downstream direction, with clearly lower amplitudes at observers B and D. At BPF, the emission of the tower shows the same directivity, yet less pronounced, whereas the directional differences at higher harmonics of BPF are marginal. The SPL increase in the plane of rotation for frequencies below BPF is mainly caused by the tower emission. This is similar to the increase of amplitudes in the tower base load spectra for $F_y$ and $M_x$ caused by pressure fluctuations on the tower surface which was described in the previous section. Thus SPL increase at frequencies below BPF is very likely induced by surface pressure fluctuations due to vortex shedding at the tower, too. Looking at the noise carpet for the third BPF harmonic in Figure 14 gives more insight into the directivity. The rotor emission is strongly directed towards $20°$ and $190°$, whereas for the tower emission only a small shift of the generally concentric shape towards $220°$ is present. The superposed signal shows a directivity towards $180°/350°$ and is slightly biased upstream. The result also shows that the shape of the SPL isolines beyond approx. $500\,\text{m}$ radius around the turbine is independent of the radius. The same behaviour can be observed for the other harmonics of BPF. Thus, the previously regarded observers at $1000\,\text{m}$ radius are clearly out of near field effects for BPF harmonics.



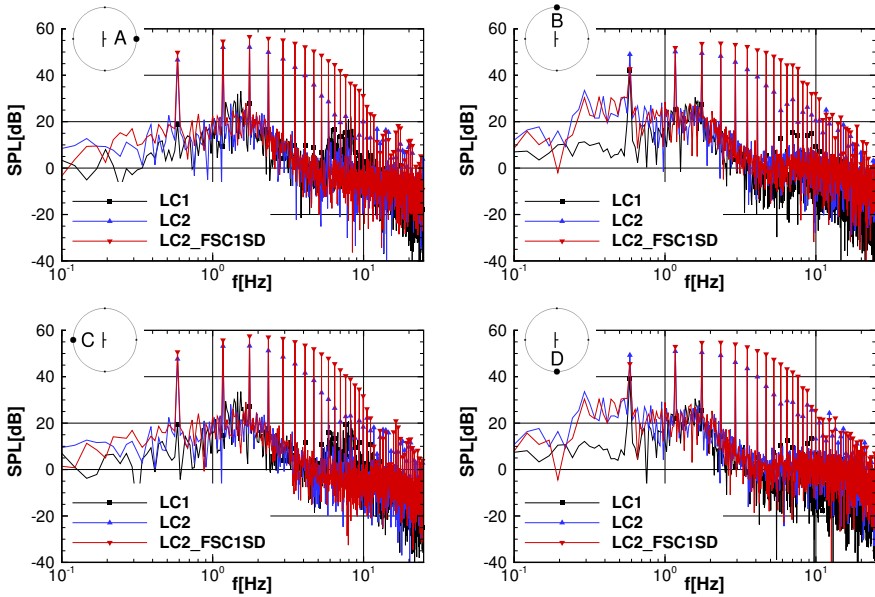

**Figure 12.** Spectra of unweighted SPL (reference sound pressure of $20\,\mu$Pa) at 4 observer positions on the ground with a distance of $1000\,$m to the turbine for cases LC1, LC2 and LC2_FSC1SD.

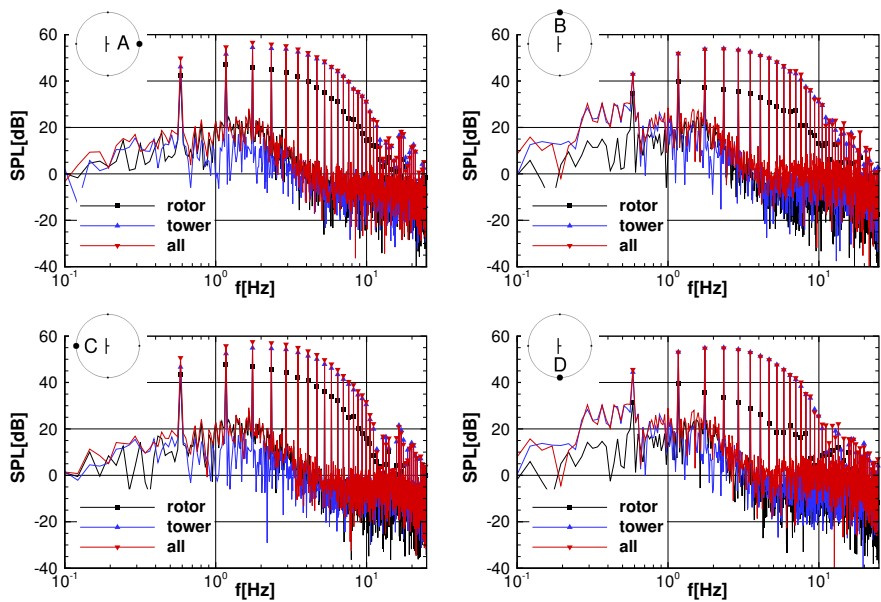

**Figure 13.** Spectra of unweighted SPL (reference sound pressure of $20\,\mu$Pa) at 4 observer positions on the ground with a distance of $1000\,$m to the turbine for case LC2_FSC1SD. Comparison of emission from rotor, tower and all surfaces.





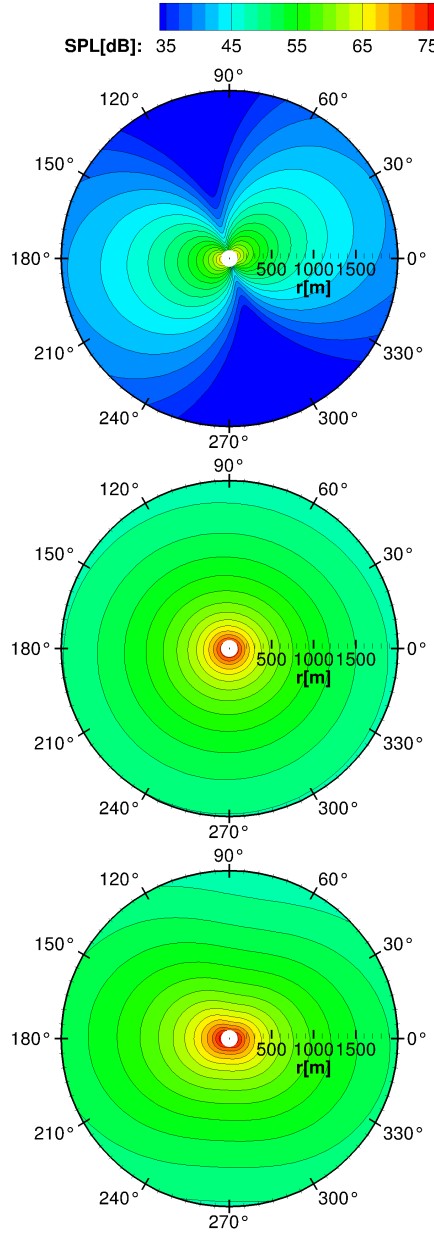

**Figure 14.** Unweighted SPL (reference sound pressure of $20\,\mu$Pa) at third BPF harmonic ($1.755\,$Hz) on ground around the turbine for case LC2_FSC1SD. Aeroacoustic emission from rotor (top), tower (middle) and all surfaces (bottom). $\Delta$SPL between black contour lines is $2\,$dB.

## 3.2 Influence of degrees of freedom at uniform inflow

In the second study the cases LC2_FSC1SD, LC2_FSC1 and LC2_FSC3 are regarded. The aim is to evaluate the influence of the degrees of freedom of the structural model on the low-frequency emissions from the wind turbine. Case LC2_FSC1SD has





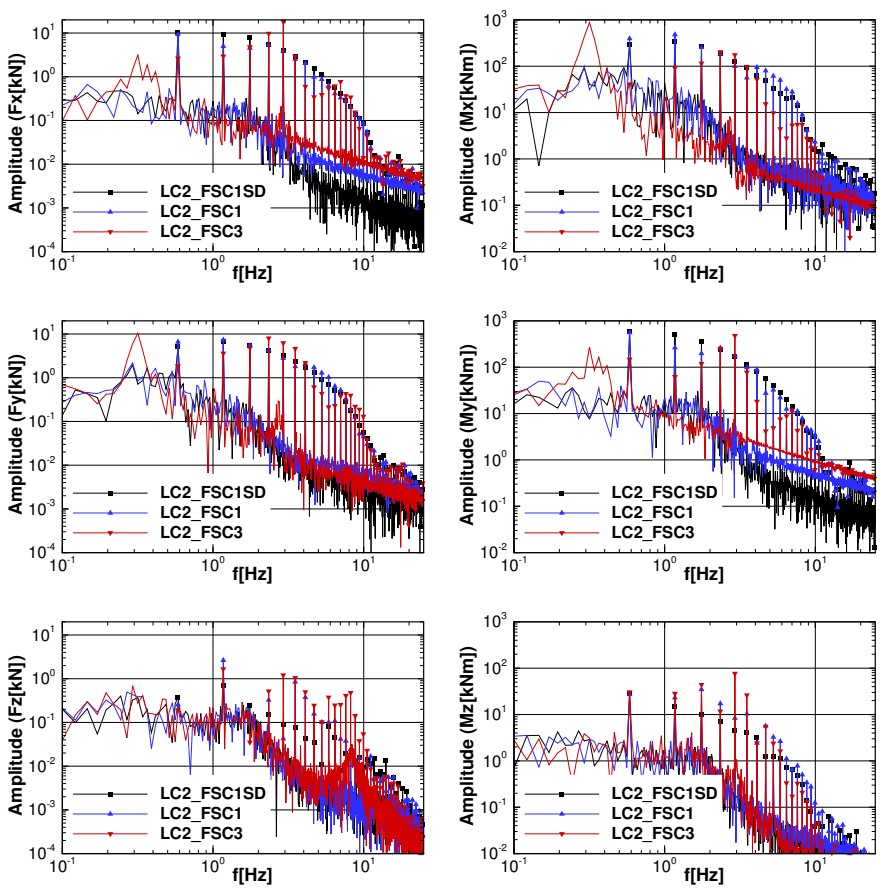

**Figure 15.** Spectra of tower base loads for the cases LC2_FSC1SD, LC2_FSC1 and LC2_FSC3.

zero degrees of freedom but considers the mean blade deformation of case LC2_FSC1 where only the rotor blades are flexible, thus it has been chosen as reference case for this study.

### 3.2.1  Tower base loads

The spectra of the tower base loads for all three cases are plotted in Figure 15. The flexibility of the rotor blades in case
5  LC2_FSC1 has mainly an impact on the amplitudes at harmonics of BPF. While the amplitudes in $F_x$ decrease slightly, the amplitudes in $F_y$ rather increase. A clear increase of $F_z$ at BPF and some higher harmonics is present. $M_x$ amplitudes also increase with the highest peaks at first and second harmonic of BPF rising by more than $30\%$ compared to case LC2_FSC1SD. On the contrary a decrease is observed for $M_y$, especially for the second and third harmonic of BPF. $M_z$ stays on a much lower level than $M_x$ and $M_y$ but amplitudes at most higher harmonics of BPF are increased compared to the reference case.
10  There are two effects which go hand in hand both having an influence on the tower base loads. By setting the blades flexible,



on the one hand, gravitational forces and inertial forces start acting and on the other hand, aerodynamic forces change due to unsteady deflection of the blades. The mean blade tip deflection applied in case LC2_FSC1SD is $6.34\,\mathrm{m}$ out of plane (OOP) and $-0.58\,\mathrm{m}$ in plane (IP). In case LC2_FSC1 the OOP deflection reaches its maximum of approximately $6.46\,\mathrm{m}$ when the blade is passing the tower, just before the blade deformation is reduced due to the tower blockage. The IP deflection oscillates

between $-0.13\,\mathrm{m}$ and $-1.02\,\mathrm{m}$, which is mainly caused by the gravitational force that makes the blade bend downwards. Due to the inertia of the blade, the IP blade tip velocity reaches its maximum just after the tower passage. This increases the absolute velocity of the blade when passing the tower and the relative flow velocity on the blade. On the other hand, the swinging of the blades mainly induces structural forces in $y$ and $z$ direction which explains the increase of $F_y$ and $F_z$ amplitudes at BPF. The enabled flexibility of the tower in case LC2_FSC3 shows a much stronger impact on the tower base loads compared to

case LC2_FSC1 as it significantly changes the structural eigenmodes of the turbine. Regarding the dominant loads $F_x$, $F_y$, $M_x$ and $M_y$, the amplitudes at first, second and third harmonics of BPF are clearly reduced. Especially the reduction at BPF is remarkable, over $70\%$ for all four load components. For $M_x$ the amplitude at BPF even drops to the level of the broadband fluctuations of the other two cases. For $F_x$ and $M_y$ the maximum amplitude shifts to the fifth harmonic of BPF which is close to three structural eigenfrequencies of the turbine. For $F_y$ and $M_x$ it occurs at approximately $0.32\,\mathrm{Hz}$ which matches with the

first side-side bending mode of the tower. An increase of the amplitudes in the frequency range around $0.32\,\mathrm{Hz}$ can also be observed for $F_x$ and $M_y$, yet less pronounced. The first fore-aft bending mode is also at this frequency but the aerodynamic damping is much higher compared to the side-side direction. For $F_z$ amplitudes at higher harmonics of BPF are increased but the maximum amplitude, occurring at BPF, is reduced. For $M_z$ a further increase of the amplitudes of second to sixth harmonics of BPF is present and the maximum amplitude is increased and shifted to the fifth harmonic of BPF.

### 3.2.2    Aeroacoustic emission

The increase of degrees of freedom in the structural model only marginally influences the SPL at the observer positions A-D (Figure 16). The spectra at observer positions A and C show a small decrease of the amplitude at BPF while there is a small increase at second to sixth harmonics of BPF. However, observers B and D show a small increase at BPF while amplitudes of higher harmonics are almost unchanged. Generally, the effect is a bit stronger for case LC2_FSC3. These small changes

might be an impact of the slightly reduced blade-tower distance and the increased blade tip velocity when the blade passes the tower which was reported in the previous section. For frequencies below BPF, the maximum amplitude increases slightly which could be induced by the structural eigenmodes of the turbine as well as by the impact of vortex shedding at the tower.

### 3.3    Influence of inflow

In the last study the influence of inflow conditions on the tower base loads and on the aeroacoustic emission is investigated. While uniform inflow was applied for the previous studies, more realistic inflow is considered in this study. Two cases, one with vertically sheared inflow (LC3_FSC3) and one with turbulent vertically sheared inflow (LC4_FSC3) are compared to



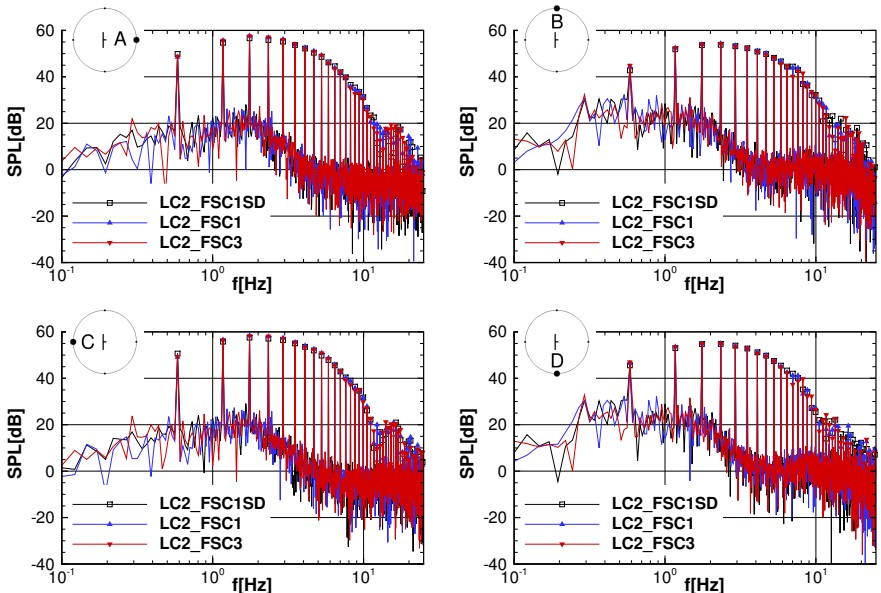

**Figure 16.** Spectra of unweighted SPL (reference sound pressure of $20\,\mu\mathrm{Pa}$) at 4 observer positions on the ground with a distance of $1000\,\mathrm{m}$ to the turbine for cases LC2_FSC1SD, LC2_FSC1 and LC2_FSC3.

the uniform inflow case (LC2_FSC3). For the turbulent inflow case a longer time series is evaluated in order to obtain more representative results.

### 3.3.1 Tower base loads

The spectra of tower base loads in Figure 17 show that for case LC3_FSC3 an increase of amplitudes is only present for $F_x$,
$F_z$, $M_y$ and $M_z$ and only at BPF. Especially the amplitude of $M_z$ at BPF rises to a remarkably high level. Amplitudes at higher harmonics of BPF tend to reduce for all loads except for $M_z$. The result also shows that the broadband load level at frequencies between first and fifth BPF harmonics rises. For $F_y$ and $M_x$ there is a clear peak just above $1\,\mathrm{Hz}$ which even exceeds the peak at BPF for $M_x$. The reduction of amplitudes at higher harmonics of BPF can be explained as a result of the reduced inflow velocity below hub height due to the power law profile. Because of the lower aerodynamic thrust in this region, OOP
deflection in front of the tower reduces to approximately $5.5\,\mathrm{m}$ compared to $6.46\,\mathrm{m}$ in case LC2_FSC3. The rise of amplitudes at BPF can be explained as an effect of vertical shear. While blade-passing is a short pulse and many higher harmonics of BPF are excited, the effect of vertical shear stretches over the whole revolution and is much closer to a sine function. Thus, the excitation of higher harmonics of BPF is much weaker compared to blade-passing. The combination of vertical shear and reduced blade-passing effect finally leads to an increase of amplitudes at BPF while amplitudes at higher harmonics decrease.
By superimposing turbulence to the vertically sheared flow in case LC4_FSC3, the character of the spectra changes as the amplitudes at BPF harmonics become much less prominent. There are some clear peaks remaining, but the broadband load





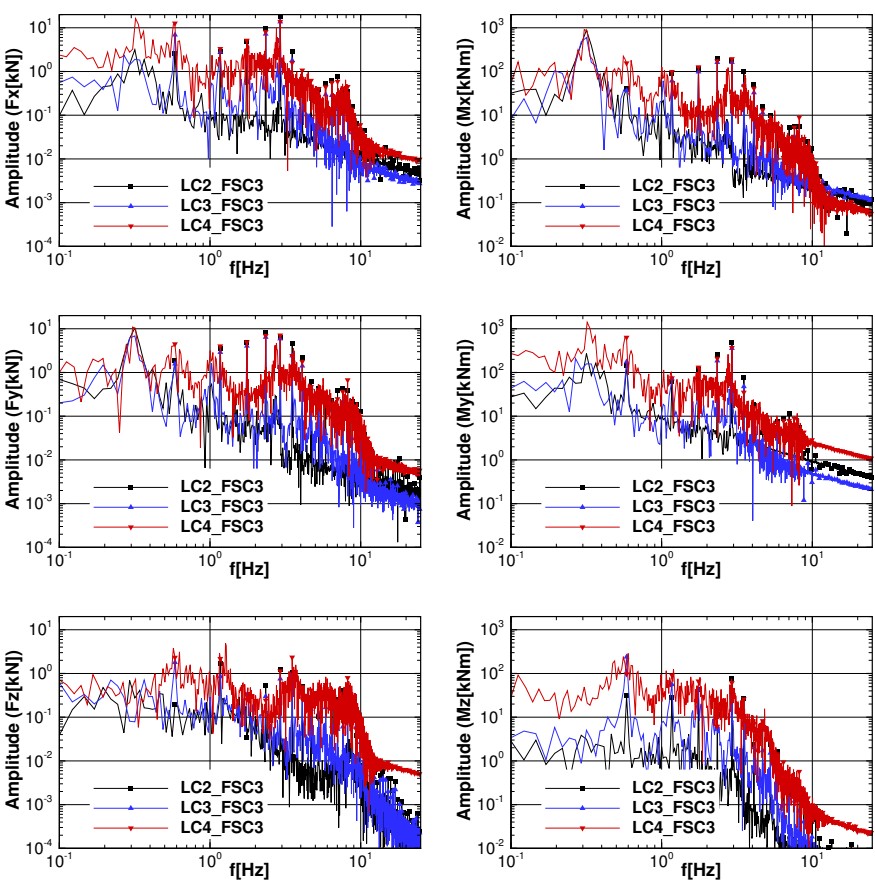

**Figure 17.** Spectra of tower base loads for the cases LC2_FSC3, LC3_FSC3 and LC4_FSC3.

level massively increases. The global maximum now arises for $M_y$ at approximately $0.32\,\mathrm{Hz}$ corresponding to an eigenmode of the structural model. Additionally the amplitude at BPF is strongly increased for $F_x$, $F_y$, $M_x$ and $M_y$; however, side peaks occur that are partially even higher. The amplitude at approximately $1\,\mathrm{Hz}$ further increases compared to case LC3_FSC3 and another wide peak appears at frequencies around approximately $2.75\,\mathrm{Hz}$, which again corresponds to nearby structural

5   eigenmodes. The higher amplitudes at frequencies near to structural eigenmodes can be explained by the broadband excitation due to the influence of turbulent inflow on the aerodynamic loads. Without turbulent inflow the main excitation occurs at BPF harmonics because all unsteady effects except for the vortex shedding are periodic with BPF (blade-tower interaction, tilt angle, vertical shear).





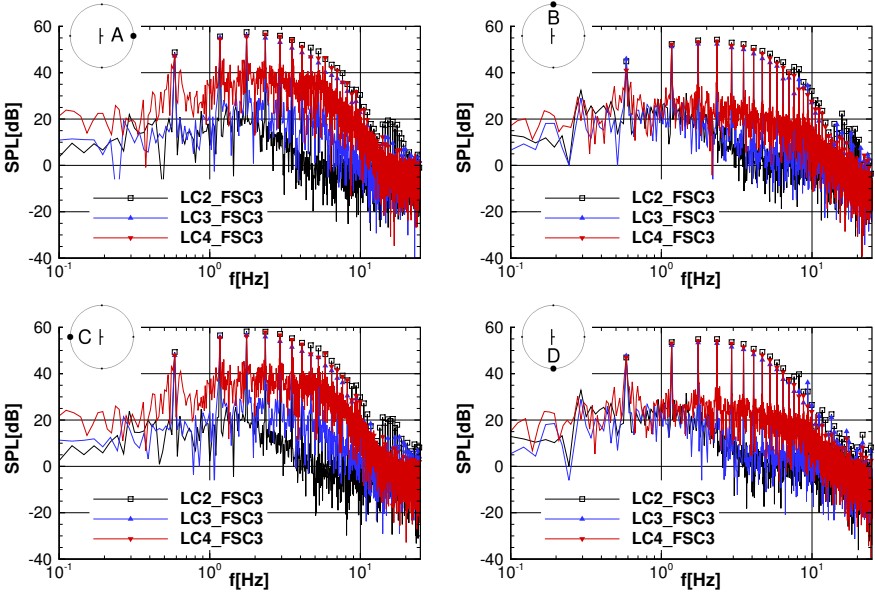

**Figure 18.** Spectra of unweighted SPL (reference sound pressure of $20\,\mu$Pa) at 4 observer positions on the ground with a distance of $1000\,$m to the turbine for cases LC2_FSC3, LC3_FSC3 and LC4_FSC3.

### 3.3.2 Aeroacoustic emissions

Figure 18 shows the spectra of the acoustic immission at observers A to D for the regarded cases. The vertically sheared inflow (case LC3_FSC3) leads to a slight decrease of SPL at BPF harmonics with a stronger effect at higher frequencies. Only a small increase of amplitude can be observed at BPF for observers B and D. For observers A and C an increase in the broadband

noise level between approximately $2\,$Hz and $10\,$Hz can be found, but it does not exceed $30\,$dB. The reduction of SPL can be explained with the reduced blade tip deflection in front of the tower already mentioned above, which reduces the pressure fluctuations on the tower. Taking the turbulent inflow into account (case LC4_FSC3) leads to an increase of the broadband noise level due to turbulent inflow noise, generated by the interaction of the rotor blade with the turbulence. The inflow noise is emitted from the rotor and predominantly directed in upstream and downstream direction, leading to higher broadband noise

levels at observers A and C compared to observers B and D. Since the rotor blades encounter the turbulence at considerably higher relative velocity than the tower, the emission from the tower hardly increases compared to case LC3_FSC3. However, despite the increased broadband noise level, the peaks at BPF harmonics are still dominant at all four observer positions.

## 4 Discussion

In the first study the influence of the presence of the tower and of steady blade deformation on low-frequency emissions was

15 evaluated at uniform inflow conditions in standalone CFD simulations. Concerning the aerodynamic loads, the presence of





the tower leads to an increase of amplitudes at BPF and its higher harmonics. Applying a steady deformation to the rotor blades further increases the amplitudes especially for higher harmonics due to the stronger blade-tower interaction. Splitting the loads up into rotor and tower loads shows that the major part of the fluctuations originates from the tower and is caused by blade-tower interaction. Load oscillations induced by vortex shedding can be observed but do not play an important role.

Evaluating the aeroacoustic immission on the ground at a distance of $1000\,\mathrm{m}$ shows similar results. Through the presence of the tower a tonal noise emission with prominent peaks at BPF harmonics arises. Reduced blade-tower distance further increases the amplitudes of BPF harmonics especially at higher frequencies. Comparing the contributions of tower and rotor to the noise emission shows a strong directivity for the rotor emission in the direction of the rotor axis and a weak directivity for the tower emission except at BPF. Generally the emission from the tower is stronger in all directions in the regarded frequency range.

This corresponds to the findings by Yauwenas et al. (2017) who did research on blade-passage noise and claimed a significant contribution of the tower. While Yauwenas et al. investigated a small model turbine with a symmetric blade in stationary air and a BPF of $45\,\mathrm{Hz}$, the present study shows that their assumption is also valid for a realistic multi megawatt turbine under uniform inflow and a BPF in the low frequency range.

In a second study, the influence of degrees of freedom in the structural model was investigated using three cases, one with
steady blade deformation already regarded in the first study, another with flexible blades and a third with additionally flexible tower and foundation. Flexible blades have only a minor impact on the calculated tower base loads. Structural eigenmodes play a more significant role in the third case when tower and foundation are flexible too. The peaks at BPF harmonics are still prominent but the amplitudes change and the maxima are shifted towards BPF harmonics close to structural eigenfrequencies. Additionally, peaks corresponding to the first bending modes of the tower ($0.32\,\mathrm{Hz}$) occur, being dominant in $F_y$ and $M_x$
spectra. Concerning aeroacoustics, the emission slightly increases but no clear influence of structural eigenmodes can be found in the regarded frequency range.

The third study deals with the influence of the inflow condition on the emissions. Uniform inflow is compared to vertically sheared inflow with and without turbulence. For vertical shear inflow tower base loads tend to increase at BPF and decrease at higher harmonics of BPF. With superimposed turbulence the peaks become much less prominent since the broadband load level
rises. Amplitudes at frequencies close to structural eigenmodes rise and BPF harmonics become less dominant in the spectra. The tonal noise level of the aeroacoustic emission tends to reduce slightly with the vertical shear and increase again due to the superimposed turbulence. The broadband noise level strongly increases especially for observers upstream and downstream of the turbine, which is mainly caused by turbulent inflow noise emitted by the rotor. Thus, the BPF harmonics become less prominent but are still dominant in the spectra.

As a generic wind turbine was investigated, no measurements for validation are available. Nevertheless, a qualitative comparison between the presented results and two studies found in literature is drawn. Zieger and Ritter (2018) showed seismic measurements in Germany that suggest an independence of discrete frequency peaks and blade-passing frequency. Although the amplitudes increase with increasing wind speed and rotational speed respectively, the frequencies of the peaks do not change. This can be interpreted as a dominance of structural eigenmodes of the turbine in the origin of the seismic waves.
However, at high (rated) rotational speed the dominant frequencies correspond very well to harmonics of the blade-passing



frequency. Saccorotti et al. (2011) analyzed seismic measurements of a gravitational wave observatory in Italy close to a wind farm and found steady spectral lines as well as time-varying peaks which could all be identified as emitted by a wind turbine. The results of both studies coincide with the findings of the presented paper where tower base loads at BPF harmonics close to eigenfrequencies of the turbine are prominent in the spectra. The tonal character of the low-frequency noise was also shown in

acoustic field measurements (Hansen et al., 2017; Pilger and Ceranna, 2017). They showed that the BPF harmonics are dominant in the measured spectra and thus the peak frequencies shift depending on the rotational speed of the turbine. Pilger and Ceranna furthermore compared measurements of a single $200kW$ turbine to estimated SPL from the Viterna method (Viterna, 1981). They found an underestimation of SPL which they explained with environmental conditions neglected in the model. Taking the present study into account it is more likely that the neglect of tower emission in the Viterna method has a major

impact on the results.

Despite the advanced modelling approach applied in the presented study, there are still several limitations that have to be mentioned. In the applied FW-H calculations effects of unsteady flow field, refraction and reflection of acoustic waves and atmospheric layering are not taken into account for the propagation. On the other hand, this makes the method very suitable for the investigation of the aeroacoustic emission of the turbine, as the immission at the observer positions is not influenced by the

effects mentioned above. Due to the computationally expensive CFD approach, there are limitations concerning the length of the time series and temporal resolution and consequently the statistical convergence of the results and the resolved frequency range. Although the flexibility of rotor blade, tower and foundation was considered in the simulations further deegrees of freedom were neglected. The drive train was kept totally rigid and at fixed rotational speed. As *SIMPACK* is a multibody solver and only deformations of points along a beam are transferred, eigenmodes of the shell cannot be considered in the presented

approach. However, the mentioned shortcomings do not not change general findings of this paper.

## 5   Conclusions

In the present paper the low-frequency emissions from a generic $5\,\text{MW}$ turbine were investigated using a high fidelity time resolved fluid-structure coupled CFD approach. Three different studies were conducted to identify sources, to better understand mechanisms and to evaluate the influence of the model complexity on the resulting emissions. Tower base loads are compared

to study the effect of structure-borne noise as seismic wave propagation cannot be calculated with the presented method. The aeroacoustic noise propagation is computed using a Ffowcs-Williams Hawkings method. To consider aeroelasticity in the simulations a new coupling of the CFD solver *FLOWer* to the MBS solver *SIMPACK* was developed and is presented in this paper. With this method not only blade deformation can be taken into account, but deformations, translations and rotations of all parts of the turbine. Thus, fluid-structure coupled simulations with flexible tower and foundation could be conducted.

As a high fidelity approach is used, the aerodynamic results are of high quality. A major advantage compared to lower fidelity approaches is that, as all geometries of the turbine are fully resolved, the unsteady pressure distributions on all surfaces, and thus all aerodynamic loads, are a direct outcome of the simulations. Regarding the aeroacoustic emission it was found that the blade-tower interaction plays a key role and the noise emitted from the tower is higher compared to the noise emitted from



the rotor. Only an indirect impact of fluid-structure-coupling on the aeroacoustics could be observed. Elastic blades reduce the distance between blade and tower and thus increase the strength of the blade-tower interaction. Turbulent inflow on the other side mainly influences the broadband noise level of the rotor. For the regarded turbulence level of $16\%$ the noise has a tonal character with dominant peaks at blade-passing frequency harmonics.

Blade-tower interaction also has a great influence on the tower base loads; however, with increasing degrees of freedom structural eigenmodes play a much stronger role than for the aeroacoustic emission and amplitudes at eigenfrequencies become more dominant when turbulent inflow is applied. Nevertheless, blade-passing frequency harmonics can still be identified in the spectra. For aerodynamic load fluctuations at uniform inflow it was found that the contribution of the tower exceeds the contribution of the rotor.

Several conclusions for the modelling of low-frequency emissions using CFD simulations can be drawn from the conducted studies. The blade-tower interaction was found to be the main source of aeroacoustic noise and triggers a major part of the aerodynamic load fluctuations. The tower itself as well as a realistic blade-tower distance has to be considered in the simulation to capture the blade-tower interaction properly. Fluid-structure coupling is the most appropriate way to a realistic blade-tower distance and is mandatory if structural emission shall be regarded. Moreover the acoustic emission from the tower has to be

considered in the noise evaluation and the loads on the tower have to be included in the fluid-structure coupling. Concerning the structural emission, not only the flexibility of the rotor blades but also of tower and foundation have to be taken into account as they change the character of the tower base load spectra. Turbulent inflow should also be taken into account, because it enhances the excitation of structural eigenmodes.

The findings can be transferred to any modelling method of low-frequency emissions from wind turbines. The method has to

be capable of capturing the impact of blade-passing not only on the blades but also on the tower and its effect on the one hand on the aerodynamic load fluctuations and on the other hand on the aeroacoustic noise emission.

Future work will deal with several of the listed limitations. A slightly smaller commercial wind turbine will be investigated numerically with the presented approach and field measurements will be available for comparison. Subsequently, the turbine

will be simulated taking into account the operational conditions of the measurements. The influence of full shell coupling on the low-frequency emission will be investigated in a future study. Based on the presented findings, constructional measures as lattice towers, increased blade tower distance or swept blades are likely to reduce low-frequency emissions and should be taken into account for future research.

*Data availability.* Data of the NREL $5\,\mathrm{MW}$ turbine is available from Jonkman et al. (2009).

*Competing interests.* The authors declare that they have no conflict of interest.



*Acknowledgements.* The studies where conducted as part of the joint research project "Objective Criteria for Seismic and Acoustic Emission of Inland Wind Turbines (TremAc), FKZ 0325839A", funded by the German Federal Ministry for Economic Affairs and Energy (BMWi). The authors are grateful for the financial support. The authors gratefully acknowledge the *High Performance Computing Center Stuttgart* for providing computational resources within the project *WEALoads*.





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
