# Peer review of "Advanced CFD-MBS coupling to assess low-frequency emissions from wind turbines"

_Wind Energy Science, 2018_

## Referee Comment (RC1) · Anonymous Referee #1 · 6 Aug 2018

The paper describes a CFD-MBS coupled method to assess tower base loads and low-frequency noise emissions of wind turbines. The topic is faced from a numerical point of view and demonstrates the applicability of the procedure to identify acoustic sources and to have a better insight on the noise generation mechanisms. The paper is relevant in the field, well written and easy to follow and includes a good and up-to-date description of the state of the art.

The numerical procedure is shown in detail and the computational setups (CFD, structural and FSI models) are well described. The results coming from three different studies are clearly presented and the main finding are well discussed. Maybe the noise results would have deserved to be presented in a more compact way in order to allow an easier comparison of the different effects on the acoustic emissions. The

work slightly suffers from the lack of method validation, yet the authors include in the article a lot of references to other works which confirm the validity of their results. The numerical procedure seems to be ready for the application to different wind turbines.

The paper is recommended for publication subject to addressing the minor revisions suggested below:

1) 2.3.2 - Mesh deformation –> The authors stated that surfaces in the CFD domain are deformed following the marker displacements. In the referee's opinion, the internal CFD domain must be deformed to follow the moving surfaces: how does the deformation library handle with this aspect? Could the author add a sentence that explains how the deformation is distributed within the CFD domain?

2) 2.3.3 - Load integration –> In the paragraph the authors wrote: "For the coupling to SIMPACK, the CFD surface is divided into segments based on the deformed marker positions. Loads are integrated for these segments and assigned to the respective markers." Could the authors explain more in detail how a CFD segment area is assigned to a single marker. Do the authors use a sort of reduction technique?

3) 2.3.4 - Communication interface –> The authors employed a typical coupling scheme between Simpack and FLOWer code, yet the two solvers run on different operating systems and the data communication must be handle by means of files. According to the referee, this strategy may lengthen the computational time due to writing and reading time. Would it be possible to run the two solvers on the same cluster exchanging information, for instance, by using an Infiniband connection? Do the authors have an idea of the time reduction in case the solvers exchange conditions by network instead of using files?

4) 2.5.2 - CFD model –> The authors show a detail of the computational grid (Figure 3), yet it would be nice if they may add a picture of the overall CFD domain. The authors mention that the fine mesh consists of 86 M of cells and a picture showing the entire domain may highlight this huge computational domain.

5) 2.5.6 - Computational approach –> Since the procedure couples an URANS solver with a multi-body code, the aeroelastic interaction (flutter) between fluid flow and moving blades should be captured. Is it correct? Could the authors clarify this aspect within the paper? Moreover, the authors stated that an artificial damping can be introduced to obtain "a fast convergence of deformation and loads to a periodic state": does this periodic state take into account the aerodamping?

6) 2.6 - Evaluation –> The referee agrees that the temporal resolution is strictly commented to the time step. Could the author add the highest frequency solved in the analyses? The author also said "To achieve the same temporal resolution in the acoustic emission, each time step a CFD surface solution was saved as input for the acoustic simulations" and all these information may require a huge amount of disk storage, how do the authors face this aspect? Finally, at the end of the paragraph the authors state that they apply FFT algorithm to the period solution: how do they check the solution periodicity?

7) 3 - Results –> The authors clearly discussed the three different studies and all the explanations are described in detail. Focusing on acoustic emissions, the authors concluded that a) the main source of noise turns out to be the blade-tower interaction, b) it is important to consider the elastic deformation which reduce the gap between blade and tower and c) the turbulence inflow only alters the broadband noise level. The authors show the noise results in term of SPL in observer positions, would it be possible to compute a PWL (sound power level) value from the results to have a global quantity describing the acoustic energy and to globally compare the different cases annoyance at a certain distance from the wind turbine?

8) In the paper the authors often write "acoustic immission". The referee thinks that is was a typo and the authors would have like to write "acoustic emissions". Please revise it in the paper.

---

## Referee Comment (RC2) · Anonymous Referee #2 · 6 Aug 2018

In the paper entitled "Advanced CFD-MBS coupling to assess low-frequency emissions from wind turbines" the intention is to evaluate the emission of aeroacoustic and seismic noise at low frequency from wind turbines. To do this, a coupling between a high-fidelity CFD RANS solver (FLOWer) and a commercial multibody solver (SIMPACK) is proposed. SIMPACK runs on a local windows machine, while FLOWer runs on a linux cluster. The communication between the two systems is done via SSH. For each timestep t, the configuration at the following step t+1 is computed by SIMPACK with loads given by FLOWer for the timestep t. From SIMPACK, at the time instant t+1 deformations that are used to modify FLOWer mesh are extracted. Now new aerodynamic loads can be computed, and the next iteration can start. Regarding aeroacoustic noise propagation, a FW-H formulation is used. Here authors use pressures and ve-

locities on all surfaces of the turbine taking into account only monopoles and dipoles. The quadropole term is neglected and this is reasonable for wind turbine applications. The presented framework is exercised on a modified NREL 5MW, operated at rated conditions.

I find the work of very good quality and to start I'd like to acknowledge the large amount of work behind the paper. The actual results are most likely not revolutionary, as on one side, these simply confirm the importance of the blade/tower interference for the low frequency emissions and on the other, given the fact that a conceptual wind turbine has been used, no validation can be performed with the current set of results. Nonetheless, the high fidelity CFD-MBS framework has been topic of development for years and it is useful to present it to the scientific community. In terms of writing, the paper is fairly well written. However, I find several paragraphs too verbose. It takes very long to go through the text and as side effect the main findings of the work do not emerge clearly. Several paragraphs look more of a technical report than from an actual scientific publication. Below I list some examples. Overall, my suggestion is to shrink the text as well as improve its readability. The paper could however be published with only a few adjustments.

First, I expose four main areas of possible improvements. After that, I list several other notes that I spotted while going through the paper. I indicate the former with page and line number.

1. Paper length

I personally find the paper too long. It takes several hours to go through it and I had to read it multiple times to capture all the aspects. In my opinion the paper has several nice findings, which however currently do not emerge clearly. Several paragraphs look more from a technical report than from an actual scientific publication. A first example consists of the way the overall goal of the work is presented. This does not stand up in the text and it is only embedded in the text at page 3-line 3. This should to

me be isolated in a well identified paragraph, so that readers (even quick readers) cannot miss it. A second example consists of paragraph 2.5.6 (with Figure 5). Does it improve readability to use almost one full page to discuss about numerical setups to decrease the CPU time? It has been certainly useful during the work, but I don't find this paragraph very useful. My suggestion is to shrink the overall paper, focusing on the strength of the computational setup and better highlighting the important findings about low-frequency emissions of WTs.

2. Comparisons

The whole section 3 is also in my opinion too long, with the focus that is more biased towards unrealistic setups (Sect. 3.1) than the realistic ones (3.3). I would consider reducing the number of comparisons, focusing on maybe 3 cases: rigid-steady state inflow, elastic-steady state inflow, elastic-turbulent. I understand that the current structure of the paper aims at distinguishing each and every single phenomenon. However I see the risk of focusing on numerical artifacts more than on actual results and realistic phenomena.

3. Appearance

The paper is generally well prepared and several nice plots help the understanding of the reader. However I suggest to eliminate some of the plots and enlarge others. Figure 1 is for example to me not needed, as well as all diagrams showing Fz and Mz. As expected, Fz and Mz never show anything interesting. Some other figures also don't add much to the discussion, see for instance Figure 10 as well as Figure 16. All plots containing the spectra could instead be enlarged to the full size of the page. Please be aware that when printed black/white all spectra are not easily readable.

4. Present vs past tense

I personally prefer papers written in present tense, while this text mixes present and past tenses, sometimes in a conflicting fashion. This does not improve readability.

Please review the text for consistency.

List of additional comments:

Page 1 line 1: I would add "wind" before turbine

Page 1 line 8: I would reformulate the sentence "The tower base loads tend to be dominated by structural eigen-frequencies with increasing complexity of the model". The sentence is not clear, and when read alone is even fairly questionable.

Page 1 line 9: Although the whole paper is about low-frequency noise, I think it would not harm to add "low-frequency" before "aeroacoustic emissions"

Page 1 line 18: I'd anticipate the verb "occur" before "in a broad frequency range"

Page 1 line 19: check the "and" and the "," in the overall sentence formulations

Page 2 line 5: "Hence" may be the wrong logical connector

Page 2 line 8: The paragraph is not well connected to the previous one

Page 2 line 30: Across the text you refer to other authors as "He" or "They". I'd prefer the passive forms for the verbs, but if you like it so, you should be consistent. Li et al. should be "They"

Page 2 line 33: "A totally new ..." may not be the right set of words to describe a coupling of existing tools within a scientific publication

Page 3 line 11: What does "strong coupling" mean?

Page 4 line 1: Tenses should all be reviewed, but here "SIMPACK is" should to me be replaced by "SIMPACK has been"

Page 4 line 5: Review the term "generally" as it is probably not the right word

Page 8 line 13: Nacelle & hub are defined as rigid body, while foundation is a rigid body connected to the ground through a spring/damper system. However in table 2 (page
9) nacelle is listed among the flexible bodies and at page 10 line 8 it is written "flexible blades as well as a flexible tower and foundation". By "non-flexible foundation" does it mean that the degrees of freedom of the spring-damper are frozen? And what about nacelle? Please clarify.

Page 8 line 14: "Details" and not "Detail"

Page 8 line 15: Here "was" is used, while a few lines later (page 9 line 2) the tense is back to present

Pag 10 line 10: In the low frequency domain the wave length is high and spectra cannot be accurately measured too close to the emitter. In the work 3600 observers are placed and the closest are only 100 m from tower base. Is the time history from those observers still accurate for the frequency band of interest? Please explain.

Page 10 line 12: Please evaluate the need to include paragraph 2.5.6

Page 12: The case LC1 is without tower and nacelle. How and where are the loads computed? Even though there is uniform inflow and no tower, shouldn't you see some periodicity in the signal due to the tilt angle?

Page 13 line 8: Please better explain the sentence "Therefore, aerodynamic loads on rotor and tower were evaluated separately." How exactly? Always at tower base?

Page 13: In figure 8, I understand the general increase of amplitudes below BPF due to shedding on the tower. Fx and My have an increase of amplitudes on the band between 5-9 Hz for LC1 and LC2. For Mz this is even more noticeable. This behavior does not appear in LC2_FSC1SD. Could you please explain what happens?

Page 14: My guess is that a Strouhal number of 0.2 was chosen as it is typical for cylinders, but it isn't mentioned. Rotor is operating at rated conditions, so let's suppose an axial induction factor of 0.33, this means that the tower experiences a flow speed of 11.3*(1-0.333)= 8 m/s. Considering the asymptotic wind speed and the average diameter, a Reynolds number around 2e6 can be calculated. Is 0.2 still a typical Strouhal

number even at such Reynolds number? Please discuss.

Page 14: 0.292 Hz should be the frequency where vortex shedding occurs. However, I don't clearly see a precise peak at this frequency. What I notice is that AROUND this frequency range there is a general increase in side-side Fy and Mx amplitudes, which makes sense because shedding frequency varies along the tower because of different diameter and inflow. Do I understand things right?

Page 24 line 30: "generic" or "conceptual" wind turbine?

Page 25 line 30: In my opinion stating that results are of "high quality" requires first a validation.

---

## Referee Comment (RC3) · Anonymous Referee #3 · 6 Aug 2018

The authors performed an exceptional study on the low-frequency emissions from a conceptual 5MW wind turbine by means of a coupled CFD-MBS approach. Simulations were performed by progressively increasing the level of complexity. This allowed to understand the influence of various aspects on the resulting acoustics emissions. Starting from a rigid simulation of the rotor-only geometry, the authors have accounted for the presence of the nacelle and tower, the flexibility of the structures and the properties of the incoming flow. The quality of the work is very good and the paper is very well presented and written. The analysis is clear, accurate and very useful for the scientific and engineering community. The numerical effort is significant and the approach adopted for the computations is good.

---

## Editor Comment (EC1) · A. Bianchini (Editor) · 10 Aug 2018

Dear authors, the Reviewers have posted their comments. The paper has received a general appreciation from them and I also confirm that the quality of your work is high. Please pay particular attention to the suggestions of Reviewers 1 and 2, especially regarding the presentation of the results and the paper length/structure. It is presently a little bit too long and verbose. This can decrease the redability of the study and the impact of the results, which are instead of high scientific quality. Best regards,

Alessandro Bianchini Associated Editore WES

---

## Author Comment (AC1) · 29 Aug 2018

**Reply to the comments of Reviewer No. 1**

Levin Klein on behalf of the authors
IAG, University of Stuttgart

August 29, 2018

The authors would like to thank the reviewer for his/her efforts and constructive comments again. They are very much appreciated and incorporated into the revised manuscript.

In this document the comments given by the 1st reviewer are addressed consecutively. The following formatting is chosen:

- The reviewer comments are marked in blue and italic.

- The reply by the authors is in black color.

- A marked-up manuscript is added. Changed sections with regard to the comments by reviewer 1 are marked in yellow.

**Minor comments "Mi"**

1. "*2.3.2 - Mesh deformation –> The authors stated that surfaces in the CFD domain are deformed following the marker displacements. In the referee's opinion, the internal CFD domain must be deformed to follow the moving surfaces: how does the deformation library handle with this aspect? Could the author add a sentence that explains how the deformation is distributed within the CFD domain?*"
The volume meshes (internal CFD domains) are deformed based on the surface mesh deformation using radial basis function. This has already been stated at the end of the section.

2. "*2.3.3 - Load integration –> In the paragraph the authors wrote: "For the coupling to SIM-PACK, the CFD surface is divided into segments based on the deformed marker positions. Loads are integrated for these segments and assigned to the respective markers." Could the authors explain more in detail how a CFD segment area is assigned to a single marker. Do the authors use a sort of reduction technique?*"

While detailed distribution of loads in form of surface pressure and friction is available in the CFD simulation, forces can only be applied at discrete positions/points in the structural model. This is explained in section 2.3.1. As written in section 2.3.3 the CFD surface is divided into segments based on the the deformed positions of these markers. Pressure and frictions are integrated over the segments and the resulting loads are assigned to the respective markers. The authors revised the section mentioned by the reviewer to make this clearer, see $\boxed{\textbf{R1:Mi2}}$ (page 5, line 148)

3. "*2.3.4 - Communication interface –> The authors employed a typical coupling scheme between Simpack and FLOWer code, yet the two solvers run on different operating systems and the data communication must be handle by means of files. According to the referee, this strategy*"

*may lengthen the computational time due to writing and reading time. Would it be possible to run the two solvers on the same cluster exchanging information, for instance, by using an Infiniband connection? Do the authors have an idea of the time reduction in case the solvers exchange conditions by network instead of using files?"*

The reading an writing time is very fast, each communication takes approx 0.1 second which is much less than 1% of the time for one time step (approx. 40 seconds) as the files are really small (approx. 22kB each in case LC2_FSC3). One big advantage of using files is, that no connection between the solvers has to be established, thus *SIMPACK* can just wait for files from *FLOWer* while *FLOWer* is restarted (due to limited job duration on clusters). Furthermore, *SIMPACK* runs only on specific Linux distributions which are not available on most clusters.

4. *"2.5.2 - CFD model –> The authors show a detail of the computational grid (Figure 3), yet it would be nice if they may add a picture of the overall CFD domain. The authors mention that the fine mesh consists of 86 M of cells and a picture showing the entire domain may highlight this huge computational domain."*

The authors understand that a figure of the computational domain would be nice, but actually it makes no sense to create such a figure including the mesh or cuts through the mesh. Compared to the size of the computational domain ($\approx$ 3 kilometre) even the resolution of the background mesh is too fine and would just result in black areas in most parts of the figure. Furthermore, the second reviewer recommended to shorten the paper and to remove less important figures.

5. *"2.6 - Evaluation –> The referee agrees that the temporal resolution is strictly commented to the time step. Could the author add the highest frequency solved in the analyses? The author also said "To achieve the same temporal resolution in the acoustic emission, each time step a CFD surface solution was saved as input for the acoustic simulations" and all these information may require a huge amount of disk storage, how do the authors face this aspect? Finally, at the end of the paragraph the authors state that they apply FFT algorithm to the period solution: how do they check the solution periodicity?"*

The authors added the Nyquist frequency as highest resolved frequency, which is know as half the sampling rate, see $\boxed{\text{R1:Mi5-a}}$ (page 11, line 284).
The surface files were only temporally on the cluster and deleted after the acoustic simulation was finished. This requires approx. 8.6 Gigabyte per revolution which can easily be handled with the available resources. The authors added "temporally" to the sentence cited by the reviewer, see $\boxed{\text{R1:Mi5-b}}$ (page 11, line 273)
Most effects on the turbine are periodic to or occur periodically with the rotation frequency or the blade passing frequency (tower passage, gravitational forces, sheared inflow). Thus, a whole-numbered number of turbine revolutions was chosen for the evaluation.

6. *"3 - Results –> The authors clearly discussed the three different studies and all the explanations are described in detail. Focusing on acoustic emissions, the authors concluded that a) the main source of noise turns out to be the blade-tower interaction, b) it is important to consider the elastic deformation which reduce the gap between blade and tower and c) the turbulence inflow only alters the broadband noise level. The authors show the noise results in term of SPL in observer positions, would it be possible to compute a PWL (sound power level) value from the results to have a global quantity describing the acoustic energy and to globally compare the different cases annoyance at a certain distance from the wind turbine?"*

PWL results from integration over a surface surrounding the acoustic source and thus is independent of the distance and yields no information about directivity and tonality. In the eyes of the authors it is not suitable to compare annoyance of the different cases which is often associated with tonal noise.

7. "*In the paper the authors often write "acoustic immission". The referee thinks that is was a typo and the authors would have like to write "acoustic emissions". Please revise it in the paper.*"

The authors also discussed this topic. They think, that immission is the right word. Emission describes everything that's emitted from the source (turbine). At a specific observer position only the immission can be measured. The approach in the paper is to compare the immissions at the observer positions and draw a conclusion on how the emission of the turbine change.

[revised manuscript text omitted]

22, 105–122, 2018.

---

## Author Comment (AC2) · 29 Aug 2018

**Reply to the comments of Reviewer No. 2**

Levin Klein on behalf of the authors
IAG, University of Stuttgart

August 29, 2018

The authors would like to thank the reviewer for his/her efforts and constructive comments again. They are very much appreciated and incorporated into the revised manuscript.

In this document the comments given by the 2nd reviewer are addressed consecutively. The following formatting is chosen:

- The reviewer comments are marked in blue and italic.

- The reply by the authors is in black.

- A marked-up manuscript is added. Changed sections with regard to the comments by reviewer 2 are marked in orange.

**General comments "C"**

1. "*Paper length: I personally find the paper too long. It takes several hours to go through it and I had to read it multiple times to capture all the aspects. In my opinion the paper has several nice findings, which however currently do not emerge clearly. Several paragraphs look more from a technical report than from an actual scientific publication. A first example consists of the way the overall goal of the work is presented. This does not stand up in the text and it is only embedded in the text at page 3-line 3. This should to me be isolated in a well identified paragraph, so that readers (even quick readers) cannot miss it. A second example consists of paragraph 2.5.6 (with Figure 5). Does it improve readability to use almost one full page to discuss about numerical setups to decrease the CPU time? It has been certainly useful during the work, but I don't find this paragraph very useful. My suggestion is to shrink the overall paper, focusing on the strength of the computational setup and better highlighting the important findings about low-frequency emissions of WTs.*"
The authors splitted the section Introduction in several subsections to improve readability and to better highlight the aims of the paper, see R2:C1-a (page 1, line 20), R2:C1-b (page 2, line 44) and R2:C1-c (page 3, line 69).
The authors agree with the reviewer that the discussion about the numerical setup is not relevant for the understanding of the paper. They fully removed section 2.5.6 (Computational approach), see R2:C1-e (page 11, line 265).

2. "*Comparisons: The whole section 3 is also in my opinion too long, with the focus that is more biased towards unrealistic setups (Sect. 3.1) than the realistic ones (3.3). I would consider reducing the number of comparisons, focusing on maybe 3 cases: rigid-steady state inflow, elastic-steady state inflow, elastic-turbulent. I understand that the current structure of*"

*the paper aims at distinguishing each and every single phenomenon. However I see the risk of focusing on numerical artifacts more than on actual results and realistic phenomena."*

The authors agree that some setups are relatively unrealistic. But a main task was to evaluate how the complexity of the setup changes the results. Many of the conclusions can not be drawn when numbers of setups is reduced. In CFD simulation it is often not possible to do a coupled simulation, because there is no structural model or even no fluid-structure coupling available. So it is quite important to see which effects can be captured without FSI and which not, and how big the difference might be.

3. *"Appearance The paper is generally well prepared and several nice plots help the understanding of the reader. However I suggest to eliminate some of the plots and enlarge others. Figure 1 is for example to me not needed, as well as all diagrams showing Fz and Mz. As expected, Fz and Mz never show anything interesting. Some other figures also don't add much to the discussion, see for instance Figure 10 as well as Figure 16. All plots containing the spectra could instead be enlarged to the full size of the page. Please be aware that when printed black/white all spectra are not easily readable."*

As suggested, the authors removed Figure 1 and 10 and the corresponding text from the paper. Figure 16 was also removed, but leaving the description of the results in the text.

To improve the readability and shorten the paper the authors removed $F_z$ and $M_z$ from the tower base load section as suggested. As $F_x$ and $M_y$ as well as $F_y$ and $M_x$ show very similar behaviour, the authors decided to focus on the bending moments $M_x$ and $M_y$ and removed the forces from the paper.

In the evaluation of the acoustic results observers A and B were removed from the figures as the behaviour of Observer C is very similar to observer A and the same applies to observers B and D. They are still mentioned in the text in connection with directivity to emphasize the similarity/symmetry.

Only a few adjustments had to be made in the text, most are just deletions (see R2:C3-a (page 5, line 140) to R2:C3-aw (page 24, line 473))

The remaining figures of tower base loads and observer spectra were enlarged to the width of on column in the final paper (before 6cm, now 8cm).

4. *"Present vs past tense: I personally prefer papers written in present tense, while this text mixes present and past tenses, sometimes in a conflicting fashion. This does not improve readability. Please review the text for consistency."*

The authors agree that the tenses are not consistent and revised the whole paper, switching past tense to present tense where reasonable, R2:C4-a (page 3, line 79) to R2:C4-aj (page 11, line 283).

**Additional comments "AC"**

1. *"Page 1 line 1: I would add "wind" before turbine "*

Added "wind" ( R2:AC1 (page 1, line 1)).

2. *"Page 1 line 8: I would reformulate the sentence "The tower base loads tend to be dominated by structural eigenfrequencies with increasing complexity of the model". The sentence is not clear, and when read alone is even fairly questionable."*

The authors reformulated the sentence, see R2:AC2 (page 1, line 9).

3. "*Page 1 line 9: Although the whole paper is about low-frequency noise, I think it would not harm to add "low-frequency" before "aeroacoustic emissions"*"

Added "low-frequency"( R2:AC3  (page 1, line 11)).

4. "*Page 1 line 18: I'd anticipate the verb "occur" before "in a broad frequency range"* "

Changed it, see  R2:AC4  (page 1, line 22).

5. "*Page 1 line 19: check the "and" and the "," in the overall sentence formulations*"

R2:AC5  (page 1, line 23)

6. "*Page 2 line 5: "Hence" may be the wrong logical connector*"

Rewrote the previous sentence to make it clearer ( R2:AC6  (page 2, line 35)).

7. "*Page 2 line 8: The paragraph is not well connected to the previous one*"

Added a sentence for connection, see  R2:AC7  (page 2, line 39).

8. "*Page 2 line 30: Across the text you refer to other authors as "He" or "They". I'd prefer the passive forms for the verbs, but if you like it so, you should be consistent. Li et al. should be "They"*"

Changed "he" and "his" to "they" and "their", see  R2:AC8  (page 3, line 65).

9. "*Page 2 line 33: "A totally new ..." may not be the right set of words to describe a coupling of existing tools within a scientific publication*"

Changed "new" to "revised" ( R2:AC9  (page 3, line 70)).

10. "*Page 3 line 11: What does "strong coupling" mean?*"

The authors removed "strong" ( R2:AC10  (page 3, line 84)), as, in their eyes, "strong coupling" is actually not clearly defined in literature. The authors originally wanted to describe "time accurate and two-way coupling" with "strong" which is described later on in more detail without using the word "strong".

11. "*Page 4 line 1: Tenses should all be reviewed, but here "SIMPACK is" should to me be replaced by "SIMPACK has been"*"

Changed it, see  R2:AC11  (page 4, line 103).

12. "*Page 8 line 13: Nacelle  hub are defined as rigid body, while foundation is a rigid body connected to the ground through a spring/damper system. However in table 2 (page 9) nacelle is listed among the flexible bodies and at page 10 line 8 it is written "flexible blades as well as a flexible tower and foundation". By "non-flexible foundation" does it mean that the degrees of freedom of the spring-damper are frozen? And what about nacelle? Please clarify.*"

You are right, this is confusing. Removed "nacelle" from table as it is rigid and only moving with the flexible tower (Table 2 (page 10)). Yes, "non-flexible" means zero degrees of freedom.

13. "*Page 8 line 14: "Details" and not "Detail"*"

Changed it, see  R2:AC13  (page 8, line 220).

14. "*Page 8 line 15: Here "was" is used, while a few lines later (page 9 line 2) the tense is back to present*"

The authors revised the whole section and switched the tense to present ( R2:AC14-a  (page 8, line 222) to  R2:AC14-e  (page 8, line 228)).

15. "*Page 11 line 10: In the low frequency domain the wave length is high and spectra cannot be accurately measured too close to the emitter. In the work 3600 observers are placed and the closest are only 100 m from tower base. Is the time history from those observers still accurate for the frequency band of interest? Please explain.*"

It is very likely that there are near field effects at a distance of 100m. That's why spectra at the observers at 1000m are regarded and the whole carpet of observers is only used to show the directivity and that 1000m is out of near field, see section 3.1.2 (page 14).

16. "*Page 10 line 12: Please evaluate the need to include paragraph 2.5.6*"

As stated above, the authors removed the whole paragraph from the paper.

17. "*Page 12: The case LC1 is without tower and nacelle. How and where are the loads computed? Even though there is uniform inflow and no tower, shouldn't you see some periodicity in the signal due to the tilt angle?*"

As stated in the text, only aerodynamic loads, calculated with respect to the tower base coordinate system, are compared. Added "(moment reference point)" for better understanding $\boxed{\text{R2:AC17-a}}$ (page 13, line 295) and adopted caption of Figure 6 (page 13). At blade passing frequency there actually is a small peak in $F_x$ and $M_y$ and a more prominent one for $M_z$. Obviously the impact on $M_x$, $F_y$ and $F_z$ is very low.

18. "*Page 13 line 8: Please better explain the sentence "Therefore, aerodynamic loads on rotor and tower were evaluated separately." How exactly? Always at tower base?*"

Aerodynamics loads from CFD simulations are obtained from integration over surfaces as described in paragraph 2.3.3 (page 5). This has just been done separately for the surfaces of the rotor and the tower. Adopted the text and caption of figure for better understanding ($\boxed{\text{R2:AC18-a}}$ (page 14, line 306), $\boxed{\text{R2:AC18-b}}$ (page 14, line 307) and Figure 7 (page 15)).

19. "*Page 13: In figure 8, I understand the general increase of amplitudes below BPF due to shedding on the tower. Fx and My have an increase of amplitudes on the band between 5-9 Hz for LC1 and LC2. For Mz this is even more noticeable. This behavior does not appear in LC2_FSC1SD. Could you please explain what happens?*"

Removed Diagram with $M_z$ from Figure as suggested by reviewer. In $M_y$ increased amplitudes at 5-9Hz are on a very low level ($\approx 0.2\%$ of maximum amplitude in case LC2).

20. "*Page 14: My guess is that a Strouhal number of 0.2 was chosen as it is typical for cylinders, but it isn't mentioned. Rotor is operating at rated conditions, so let's suppose an axial induction factor of 0.33, this means that the tower experiences a flow speed of 11.3\*(1-0.333)= 8 m/s. Considering the asymptotic wind speed and the average diameter, a Reynolds number around 2e6 can be calculated. Is 0.2 still a typical Strouhal number even at such Reynolds number? Please discuss.*"

The authors adjusted the Strouhal number to 0.24 which better fits the high Reynolds number. Adopted the text accordingly ($\boxed{\text{R2:AC20}}$ (page 14, line 315)).

21. "*Page 14: 0.292 Hz should be the frequency where vortex shedding occurs. However, I don't clearly see a precise peak at this frequency. What I notice is that AROUND this frequency range there is a general increase in side-side Fy and Mx amplitudes, which makes sense because shedding frequency varies along the tower because of different diameter and inflow. Do I understand things right?*"

Yes, that's how the authors understand it too. As the highest peak in this frequency range is at 0.292 Hz it was chosen for the evaluation of surface pressure. For sure there is a general increase in the frequency range around this frequency. Adopted text for better understanding, see R2:AC21 (page 14, line 321).

22. *"Page 24 line 30: "generic" or "conceptual" wind turbine?"*

"Generic" is widely used in the context of the NREL 5MW turbine which is the basis of the investigated turbine.

23. *"Page 25 line 30: In my opinion stating that results are of "high quality" requires first a validation."*

The capability of the CFD code for wind turbine simulations has been proven in several projects. E.g the European Avatar Project. A validation in the actual case is actually not possible. Nevertheless, the authors removed the whole sentence from the paper, (R2:AC23 (page 25, line 518)).

[revised manuscript text omitted]

22, 105–122, 2018.

---

## Author Comment (AC3) · 29 Aug 2018

Dear Reviewer, thank you very much for your comment. Kind regards, Levin Klein

---

## Editor Comment (EC2) · A. Bianchini (Editor) · 30 Aug 2018

Dear authors, I have gone through your responses to Reviewer's comment. Please upload a revised version of the manuscript. With respect to those used during the response to Reviewers, please provide a "clean" version of the same, only highlighting the new sections with a different color. Looking forward to receiving your paper, best regards

---

## Referee Report (RR1)

Referee 1 – Comments to the revised paper

The paper is focused on a numerical method to assess tower base loads and low-frequency noise emissions of wind turbines. The computational setups is clearly described and the result are well presented confirming the credibility of the work and its relevance in the field.

The reviewer noticed that all the requested "minor revisions" were fulfilled by the authors by correctly answer the reviewer's questions and adding further explanation in the paper. So, the paper can be considered acceptable as is.

Concerning the last reviewer's request about the "acoustic immission" definition, the reviewer got the point (thanks for the clear explanation), so the use of this archaic English adjective seems to be fully justified.

Finally the reviewer noticed that in the "version 2" of the paper, some inconsistencies seem to be present in the data included in Fig. 9, 10 and 14 (there inconsistencies were not present in the previous version). Please be careful when submitting the final version.

---

## Author Response (AR2)

**Reply to the comments of the Reviewers**

Levin Klein on behalf of the authors
IAG, University of Stuttgart

September 19, 2018

The authors would like to thank the reviewers for their efforts and constructive comments again. They are very much appreciated and incorporated into the revised manuscript.

As no comments of the 1st reviewer have to be answered, in this document only the comments given by the 2nd reviewer are addressed. The following formatting is chosen:

- The reviewer comments are marked in blue and italic.

- The reply by the authors is in black.

- A marked-up manuscript is added. Deleted sections are crossed out in red, added sections are marked in blue. As all changes in text are subject to the first comment or correction of typos, there is no additional marking.

**General comments**

1. *"However, before publication I suggest to carefully review once more the tense of the verbs throughout the text, as present and past tenses are still mixed. I believe that the paper readability would improve by simply switch all verbs describing the author's activities to present (why are the abstract and the conclusions using both present and past tenses? isn't present ok?), leaving the past tense only to referencing existing works."*
We agree with the reviewer and have to admit that usage of tenses was still not consistent. We revised the tenses in the paper again following his suggestion. In the wake of this we had to revise a few sentences.

2. *"Figures 1-2-4: they are to me too small and not easily readable, I'd enlarge them"*
We enlarged the mentioned figures.

3. *"Table 1: I'd use appropriate u.o.m. (tons, MNm, ...), but it might be a matter of taste"*
We would like to keep the units as they are. Data in SI units are used by most programs and one can just copy & paste the data to build up a foundation model. In the eyes of the authors, except mass, most of the values are hard to grasp and this won't get better by replacing e6 by M or e9 by G.

4. *"Figures 6-7-9-10-12-13-14: the two plots may align on the same line"*
We know that it seems unreasonable to have the figures on top of each other in the one column manuscript layout but they actually better fit into the final two column layout of the paper. The authors tried to align them in the same line as two-column figures but then two figures won't be on one side any more, moving them farer away from their actual appearance in the the text.

List of the relevant changes in the document (numbers refer to second submission):

- Revised tenses in the whole paper
- Corrected a few typos
- Enlarged figures 1, 2 and 4
- Added the correct diagrams in figures 9, 10 and 14  that were accidentally wrong in the first revision but correct in the first submission.

[revised manuscript text omitted]